# Transcranial focused ultrasound stimulation enhances semantic memory by modulating brain morphology, neurochemistry and neural dynamics

JeYoung Jung [1,2,3,4,5] ✉, Cyril Atkinson-Clement [1,2,3], Marcus Kaiser [2,3,6] & Matthew A. Lambon Ralph [7]

The ventromedial anterior temporal lobe (ATL) is a core transmodal hub for semantic memory, yet non-invasive modulation of this region has remained challenging. Transcranial ultrasound stimulation (TUS) offers high spatial precision suitable for deep brain targets. In this study, we investigated whether theta-burst TUS (tbTUS) to the ventromedial ATL enhances semantic memory, using a multimodal neuroimaging approach—magnetic resonance spectroscopy (MRS), functional MRI (fMRI), and voxel-based morphometry (VBM). Compared to control stimulation, tbTUS improved semantic task performance. MRS showed decreased GABA and increased Glx, reflecting shifts in excitation-inhibition balance, alongside increases in NAA, creatine and choline, suggesting enhanced neuronal metabolism. fMRI demonstrated reduced ATL activity during semantic processing and strengthened effective connectivity across the semantic network. VBM revealed increased ATL grey matter volume. These findings provide convergent evidence that tbTUS modulates neurochemistry, functional dynamics, and brain morphology to enhance semantic memory, highlighting its neurorehabilitation potential.

Every day, we rely on semantic memory—the repository of our collective knowledge about the world—to engage with others and navigate daily life. By enabling language comprehension and production, object recognition, and the interpretation of everyday events, semantic memory is a core component of human cognition and behaviours[1]. Impairments in semantic memory significantly affect quality of life. Such deficits are commonly observed in neurodegenerative disorders like dementia[2,3]. Thus, understanding the mechanisms underlying semantic memory is critical for addressing fundamental questions about human

cognition and developing effective interventions for conditions that disrupt it.

In the brain, the anterior temporal lobe (ATL) plays a critical role in the representation of coherent semantic representations, functioning as a cross-modal, transtemporal hub for semantic memory[4–7]. Evidence supporting it comes from studies of semantic dementia, which arises from progressive ATL-centred atrophy[8,9], as well as findings from neuroimaging techniques such as functional magnetic resonance imaging (fMRI)[10,11] and magnetoencephalography (MEG)[12,13]. Studies employing non-invasive brain stimulation techniques such as

[1]School of Psychology, University of Nottingham, Nottingham, UK. [2]Centre For Neurotechnology, Neuromodulation and Neurotherapeutics, University of Nottingham, Nottingham, UK. [3]NIHR Biomedical Research Centre, University of Nottingham, Nottingham, UK. [4]Centre for Dementia, Institute of Mental Health, University of Nottingham, Nottingham, UK. [5]Brain Convergence Research Centre, College of Medicine, Korea University, Seoul, South Korea. [6]Department of Neurosurgery, Ruijin Hospital, Shanghai Jiao Tong University School of Medicine, Shanghai, China. [7]MRC Cognition and Brain Sciences Unit, University of Cambridge, Cambridge, UK. ✉e-mail: jeyoung.jung@nottingham.ac.uk

transcranial magnetic stimulation (TMS)[14–16] and transcranial electrical stimulation (tES)[17] have further established the causal role of the ATL in semantic memory function. Specifically, cortical grid-electrode stimulation and electrocorticography studies[18–20] and distortion-corrected fMRI investigations[21,22] have estimated more precise spatial and temporal granularities within the ATL. These studies suggest that the cross-modal hub of semantic memory is centred on the ventromedial ATL[6,23].

Magnetic resonance spectroscopy (MRS) is an in vivo magnetic resonance imaging (MRI) technique that quantifies neurochemical concentrations in the brain, including GABA (gamma-aminobutyric acid, a major inhibitory neurotransmitter) and glutamate (a major excitatory neurotransmitter). Recent studies employing MRS further support the ventromedial ATL as the centre of the semantic representation hub by providing insights into its neurochemical underpinnings[24–26]. In combination with fMRI, these studies have demonstrated that GABAergic signals within the ventromedial ATL are associated with semantic task-induced fMRI activity and semantic task performance[24,26]. However, direct stimulation of the ventromedial ATL remains challenging with current non-invasive techniques such as TMS or tES.

Low-intensity transcranial ultrasound stimulation (TUS) is an emerging non-invasive brain stimulation technique that has gained recognition for its potential applications in neuroscience research and the treatment of neurological and psychiatric disorders. TUS operates by transmitting sound waves in the 100 to 1000 kHz frequency range through the skull, focusing acoustic energy on specific brain regions with millimetre-level spatial precision. Compared to TMS and tES, TUS offers several unique advantages. Compared to TMS and TES, it is safe, painless and capable of precisely targeting deep brain structures, producing neuromodulatory effects that can last from milliseconds to hours after stimulation[27–30]. These prolonged effects are particularly important as they may induce neuroplastic changes, such as long-term potentiation (LTP) or long-term depression (LTD), making TUS a promising tool for therapeutic applications by modulating of neural activity in specific regions or networks[30,31]. TUS primarily induces neuromodulation through the mechanical interaction of ultrasound waves with cells in the targeted brain areas[32,33]. However, the neuromodulatory mechanisms of TUS and its effects on the brain and cognition are still not well understood.

Recent studies have used TUS in combination with MRI to investigate its effect on neurochemistry and functional brain networks. fMRI studies have demonstrated that TUS induced both local and widespread changes in brain activity and functional connectivity. These effects have been reported in both macaques[34–37] and humans[38–42]. When combined with MRS enables the assessment of neurochemical changes induced by TUS in targeted brain regions. To date, only two TUS studies have been shown to reduce GABA levels in the posterior cingulate cortex (PCC)[39] and motor cortex[43], while also increasing glx (glutamine + glutamate) in the motor cortex[43]. These findings highlight TUS's potential as a neuromodulatory tool capable of modulating neurotransmitters and neural activity within functional networks. However, evidence linking these changes to human cognition remains very limited. Here, we investigate how TUS can induce lasting effects on semantic memory.

A theta-burst protocol for transcranial focused ultrasound stimulation (tbTUS) has often been associated with excitatory effects[43–45]. However, accumulating evidence indicates that tbTUS can produce both facilitatory and inhibitory outcomes depending on stimulation parameters, brain state, and experimental context[44,46]. A recent tbTUS study[47] has been shown to induce excitatory neuromodulatory effects, enhancing cortical excitability in the motor cortex and improving motor task performance. Furthermore, a study using MRS and fMRI has demonstrated that tbTUS decreases regional GABA concentrations in targeted areas while increasing functional connectivity in functionally-connected remote regions[39]. Building on these findings,

we aimed to enhance semantic memory performance by delivering tbTUS to the ventromedial ATL, compared to a validated control condition, the ventricle TUS[48]. Our study combined TUS, MRS and fMRI to explore the effects of tbTUS on semantic task performance, neurochemistry, cortical activity and effective connectivity within the semantic network during semantic processing. While both ATLs contribute to semantic memory[6], we selected the left ATL as our target region. This decision is supported by rTMS findings demonstrating that stimulation of either ATL yields comparable effects on semantic processing[14,15,49]. Specifically, we hypothesised that tbTUS applied to the ATL would decrease GABA concentrations and increase glx levels in the ATL. Second, tbTUS would induce fMRI blood-oxygen-level-dependent (BOLD) signal changes during semantic tasks, both in the ATL and other regions associated with semantic processing, including the inferior frontal gyrus (IFG) and posterior middle temporal gyrus (pMTG)[6]. Third, if tbTUS-induced modulation of the ATL reflects adaptive short-term plasticity within the semantic network, effective connectivity (causal and directional inference between regions) would reveal increased facilitatory connectivity between the ATL and other semantic regions, leading to improved task performance. In addition, given animal studies revealing structural changes following TUS[50,51], we examined potential structural changes in the ATL to assess whether similar effects occur in humans. This comprehensive approach aimed to advance our understanding of how TUS can modulate the ventromedial ATL, influence the wider semantic network, and enhance cognitive performance.

## Results

We investigated the regional and task-specific effects of tbTUS on semantic memory, focusing on structural, biochemical, functional and behavioural changes. Participants completed three sessions (Fig. 1). In session 1, a high-resolution T1-weighted MR image was acquired for neuronavigation. In sessions 2 and 3, tbTUS was applied to either the left ventromedial ATL (active stimulation) or the left lateral ventricle (control stimulation) with a 5-day gap at minimum. Following each tbTUS, participants underwent scanning. In these sessions, participants performed a semantic association task and a pattern-matching control task before and after tbTUS (Fig. 1B, see details in Methods). During structural imaging and MRS acquisition, participants were instructed to remain relaxed with their eyes open. MRS was used to detect and quantify regional GABA+ and glx (glutamate + glutamine) concentrations in the ATL and occipital cortex (OCC) as a control region (Fig. 1C). For fMRI, participants performed the semantic and control tasks in a block design (Fig. 1B). After the scanning session, participants repeated the tasks and completed an aversive response questionnaire related to tbTUS at the end of the study.

TUS was delivered using theta-burst parameters as follows: central frequency = 500 kHz, pulse duration = 20 ms, pulse repetition interval = 200 ms, duty cycle = 10%, pulse repetition frequency = 5 Hz, ISPPA = 54.51 W/cm², and total duration = 80 s (Fig. 2A). tbTUS was applied to either the ventromedial ATL or the ventricle in the left hemisphere. The ATL target coordinates (MNI: −36, −15, −30) were selected based on prior studies[24,52]. The ventricle target was manually identified as the centre of the left lateral posterior ventricle using each participant's T1-weighted image[48]. Acoustic simulations confirmed precise targeting of both the ventromedial ATL and ventricle (Fig. 2B). The transcranial mechanical index (MItc; max = 1.33) and temperature rise (max at target = 0.567 °C; max at tissue = 2.88 °C: max at skull = 4.69 °C) for both regions remained below the safety limits recommended by the United States Food and Drug Administration (US FDA) (see details in Methods and Source Data).

### Effects of tbTUS on behaviour

A 2 × 2 repeated measures ANOVA was conducted with stimulation (ATL vs. ventricle) and session (PRE vs. POST) as within-subject factors

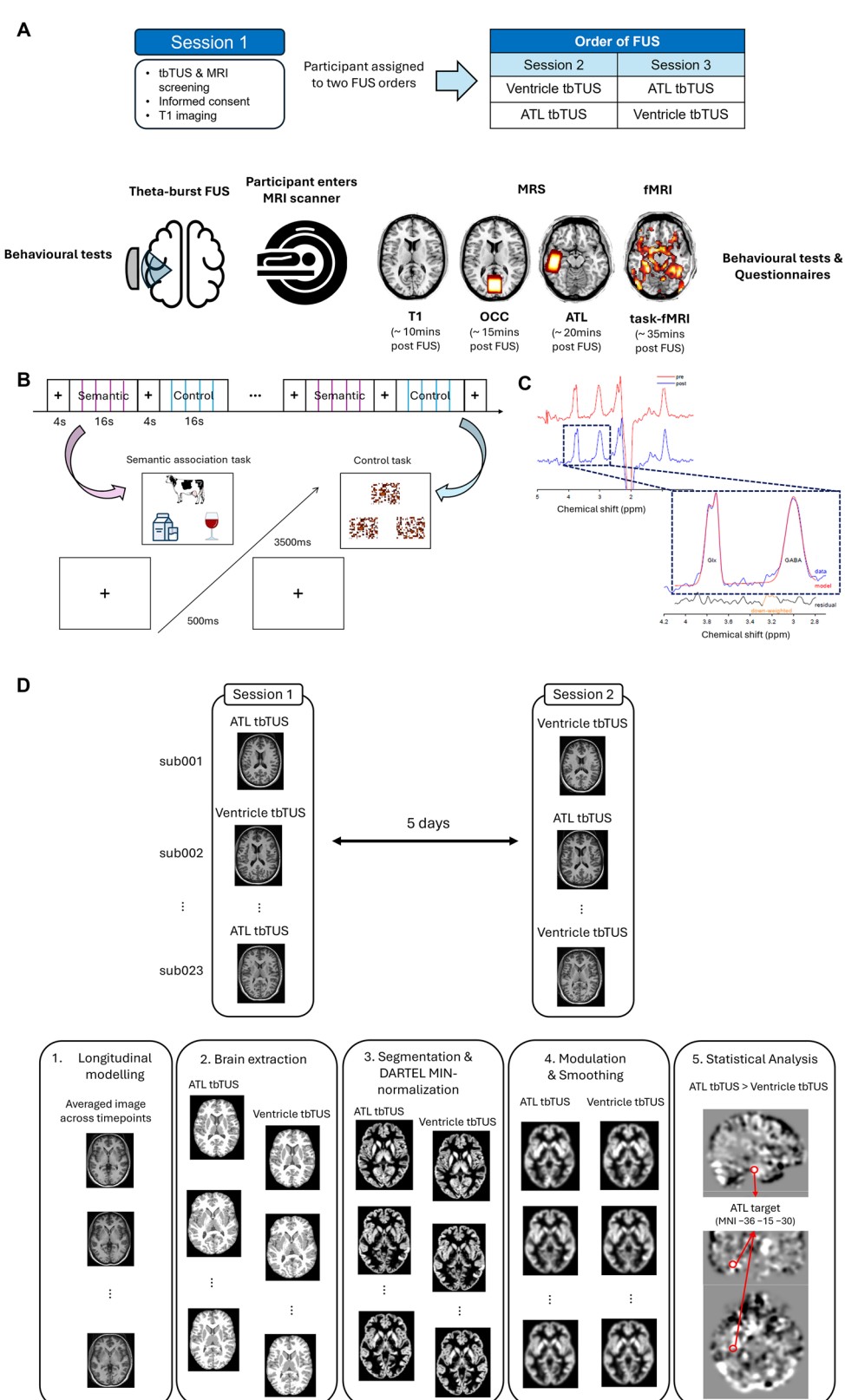

**Fig. 1 | Study design. A** Experimental design and procedure. **B** The block design of fMRI task and an example of the semantic association task (left) and the control task (right: pattern matching). Each task block consisted of four trials, and eleven blocks of each task were alternated (e.g., A-B-A-B) with a 4000 ms fixation period between blocks. Each trial begins with a fixation cross, followed by a stimulus containing three items, a target at the top and two choices at the bottom. In the semantic association task, the correct answer is "milk," as it is meaningfully related to "cow." **C** A representative MRS spectrum with estimated peaks. The red line represents the raw spectra, and the blue line shows the post-processed spectra. Within the dotted box, the red line illustrates the estimated model fit for GABA+ and Glx. The black line indicates residuals. **D** Diagram of the longitudinal voxel-based morphometry (VBM) analysis (see the "Methods", "VBM analysis"). The images used in (**A**, **B**) are either original or sourced from the public domain (copyright-free). The final figure composition was executed using Microsoft PowerPoint.

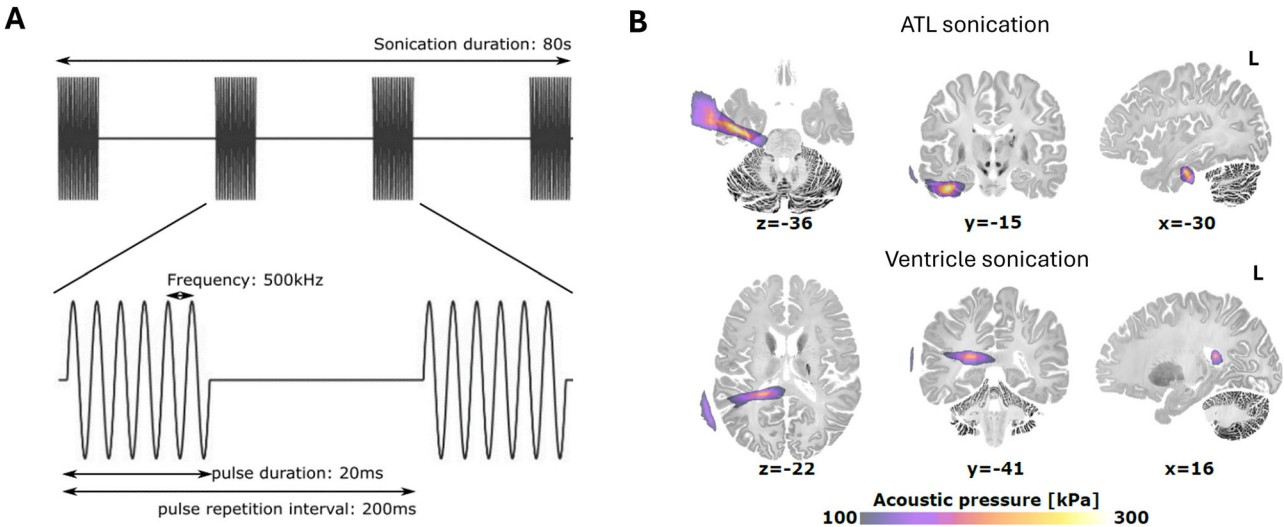

**Fig. 2 | Transcranial ultrasound stimulation. A** tbTUS protocol. **B** Simulated ultrasound pressure field averaged from all participants. Group-level results indicated a peak pressure at the target of 578.2 ± 101.3 kPa for ATL sonication, and of 624.6 ± 90 kPa for ventricle sonication.

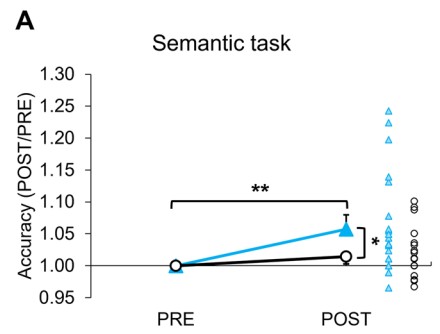

**Fig. 3 | Behavioural results. A** tbTUS-induced change in the normalised accuracy in the semantic task. Planned two-sided $t$ tests showed a significant increase for ATL stimulation from PRE to POST, $t(22) = 3.56$, $P = 0.002$, Cohen's $d = 0.74$, 95% Confidence Interval (CI) = [0.024, 0.091]. For ventricle stimulation, the PRE–POST comparison was not significant, $t(21) = 1.76$, $P = 0.093$, Cohen's $d = 0.37$, 95% CI = [−0.040, 0.048]. **B** tbTUS-induced changes in the normalised accuracy in the control (pattern matching) task. Planned two-sided $t$ tests showed no significant change for ATL stimulation from PRE to POST, $t(22) = −1.37$, $P = 0.186$, Cohen's $d = −0.28$, 95% CI = [−0.025, 0.005]. Similarly, ventricle stimulation showed no significant PRE–POST difference, $t(21) = −0.87$, $P = 0.392$, Cohen's $d = −0.18$, 95% CI = [−0.045, 0.018]. Light blue lines and triangles indicate the ATL stimulation. Black lines and circles represent the control (ventricle) stimulation. Lines represent the mean and standard error of the mean for each condition. **$P < 0.01$, *$P < 0.05$.

to evaluate the effects of tbTUS in each task. The summary of task performance is provided in the Supplementary Information, Supplementary Figs. 1–2 and Source Data.

In the semantic task, there was a significant main effect of session ($F_{1,21} = 10.61$, $P = 0.004$) and an interaction between the stimulation and session ($F_{1,21} = 5.09$, $P = 0.035$) on accuracy. There were no significant effects found in the control task ($Fs < 1.73$, $Ps > 0.202$). Planned $t$ tests were performed on normalised task performance, accounting for individual variability in baseline (pre-session) performance (Fig. S1): individual performance was divided by the pre-session performance, so the normalised pre-session was always '1'. ATL tbTUS significantly increased semantic task accuracy compared to the pre-session ($t = 3.56$, $P = 0.002$) and the control stimulation ($t = 2.46$, $P = 0.023$) (Fig. 3A). No other significant effects were found for accuracy in the control task or following control (ventricle) stimulation (Fig. 3B). For reaction time (RT), we only found a significant main effect of session in both tasks (semantic: $F_{1,21} = 88.56$, $P < 0.001$; control: $F_{1,21} = 8.56$, $P = 0.008$). Post hoc $t$ tests revealed that participant showed faster RT in their post-session regardless of stimulation or task ($ts > −3.04$, $Ps < 0.05$) (Fig. S3). There was no other significant effect found in RT.

## Effects of tbTUS on local neurochemistry

We conducted planned paired $t$ tests to assess the effects of tbTUS on regional neurochemical levels within the ATL and OCC (control) voxels of interest (VOI). In the ATL VOI, ATL tbTUS significantly reduced GABA+ ($t = -1.79$, $P = 0.043$) and increased Glx ($t = 2.60$, $P = 0.009$) compared to the control stimulation (Fig. 4A–C). In addition, ATL tbTUS led significant increases in Cho (choline, $t = 2.99$, $P = 0.008$), tCr (total creatine, $t = 2.54$, $P = 0.021$), and NAA (N-Acetylaspartate, $t = 2.56$, $P = 0.020$) (Fig. 4D–F). No significant differences in neurochemical levels were observed between ATL and control stimulation in the OCC VOI (Fig. 4G–L). All data are summarised in the Source Data.

To examine the excitation-inhibition balance (EIB), we computed Glx/GABA ratio and performed the paired $t$ test. ATL tbTUS significantly increased EIB compared to the control stimulation ($t = 7.14$, $P < 0.001$) (Fig. 5A). There was no difference between the stimulations in the OCC VOI (Fig. 5B). Importantly, the ATL EIB was significantly correlated with semantic task performance (RT: $r = 0.5$, $P = 0.03$) (Fig. 5C).

## Effects of tbTUS on local brain morphometry

We performed VBM (Voxel-Based Morphometry) analysis to explore regionally specific GM (Grey Matter) changes following ATL vs.

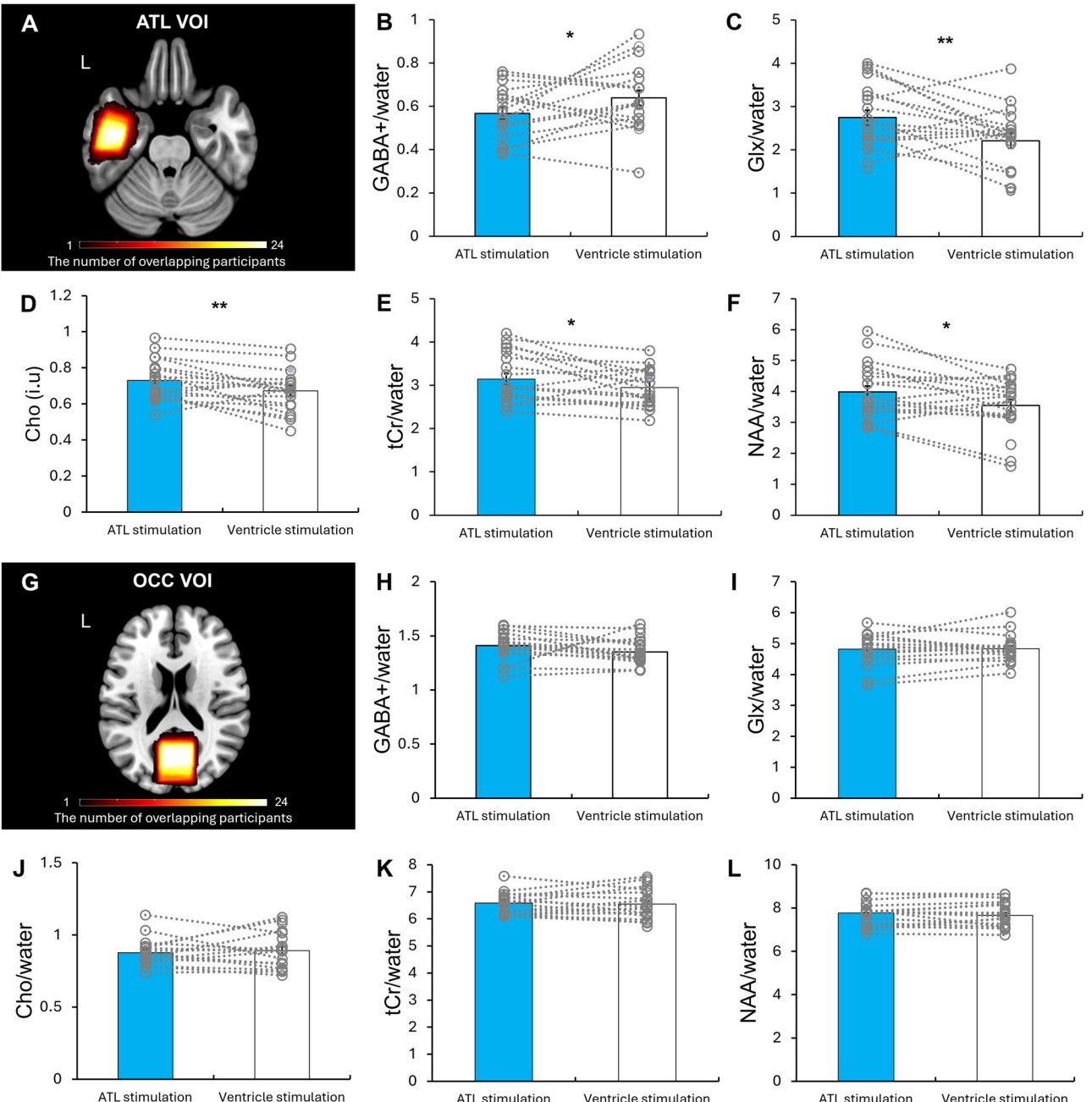

**Fig. 4 | The results of MRS analysis. A** The location of ATL VOI ($40 \times 20 \times 20$ mm³). The colour bar indicates the number of overlapping participants. **B–F** Comparison of neurochemical levels between ATL and ventricle (control) stimulation in the ATL VOI. Planned one-sided $t$ tests showed a significant decrease in GABA, $t(16) = -1.79$, $P = 0.043$, Cohen's $d = -0.43$, 95% Confidence Interval (CI) = [−0.154, 0.013] and a significant increase in Glx, $t(16) = 2.60$, $P = 0.009$, Cohen's $d = 0.63$, 95% CI = [0.093, 0.910]. Planned two-sided $t$ tests revealed significant increases in Cho, $t(17) = 2.99$, $P = 0.008$, Cohen's $d = -0.43$, 95% CI = [−0.154, 0.114]; $t$Cr, $t(17) = 2.54$, $P = 0.021$, Cohen's $d = 0.60$, 95% CI = [0.063, 0.683]; and NAA, $t(17) = 2.56$, $P = 0.020$, Cohen's $d = 0.60$, 95% CI = [0.075, 0.784]. **G**) The location of OCC (control) VOI ($30 \times 30 \times 30$ mm³). The colour bar indicates the number of overlapping participants. **H–L** Comparison of neurochemical levels between ATL and ventricle

(control) stimulation in the OCC (control) VOI. Planned one-sided $t$ tests showed no significant changes in GABA, $t(15) = 1.19$, $P = 0.126$, Cohen's $d = 0.29$, 95% CI = [−0.034, 0.123], or in Glx, $t(15) = -0.40$, $P = 0.346$, Cohen's $d = -0.10$, 95% CI = [−0.256, 0.175]. Planned two-sided $t$ tests also revealed no significant changes in Cho, $t(15) = -0.82$, $P = 0.424$, Cohen's $d = -0.19$, 95% CI = [−0.107, 0.047]; $t$Cr, $t(15) = -0.03$, $P = 0.974$, Cohen's $d = -0.01$, 95% CI = [−0.225, 0.218]; or NAA, $t(16) = -0.02$, $P = 0.986$, Cohen's $d = -0.004$, 95% CI = [−0.179, 0.175]. The colour bar indicates the number of overlapping participants. Light blue bars represent the ATL stimulation. White bars indicate the control (ventricle) stimulation. Grey circles represent individual data. Bars represent the mean and standard error of the mean for each condition. **$P < 0.01$, *$P < 0.05$.

ventricle stimulation, using a priori a 5 mm sphere VOI in the left ATL (MNI: −36, −15, −30). The VBM results revealed that ATL tbTUS significantly increased GM volume in the left ventromedial ATL (MNI: −38, −12, −30, $t = 3.12$, ks = 11, $P_{\text{SVC-FWE corrected}} = 0.042$) compared to the control stimulation (Fig. 6A). No significant GM changes were observed in the comparison of ATL stimulation <control stimulation.

## Effects of tbTUS on local and remote activation within the semantic network

fMRI results revealed that the semantic task evoked increased activation in the ATL, prefrontal and posterior temporal cortex relative to the control task (Fig. 6B, Top). There were no significant differences in the whole-brain analysis when comparing ATL tbTUS with ventricle tbTUS

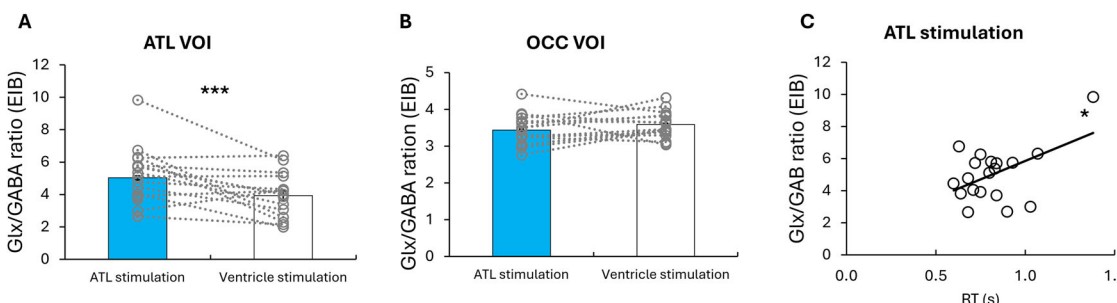

**Fig. 5 | The results of MRS analysis. A** Comparison of EIB between ATL and ventricle (control) stimulation in the ATL VOI. A two-sided paired *t* test revealed a significant increase in ATL EIB compared with ventricle stimulation, *t*(15) = 7.14, *P* < 0.001, Cohen's *d* = 1.84, 95% Confidence Interval (CI) = [1.999, 3.697]. **B** Comparison of EIB between ATL and ventricle (control) stimulation in the OCC VOI. A two-sided paired *t* test showed no change in OCC EIB, *t*(15) = 1.14, *P* = 0.271,

Cohen's *d* = 0.28, 95% CI = [−0.485, 1.608]. **C** Relationship between the ATL EIB and semantic task performance (RT) (a correlation analysis between semantic task performance and EIB following the ATL stimulation, *r*(19) = 0.5, *P* = 0.03. Grey circles represent individual data. Bars represent the mean and standard error of the mean for each condition. ***\*\*\*P* < 0.001, \**P* < 0.05.

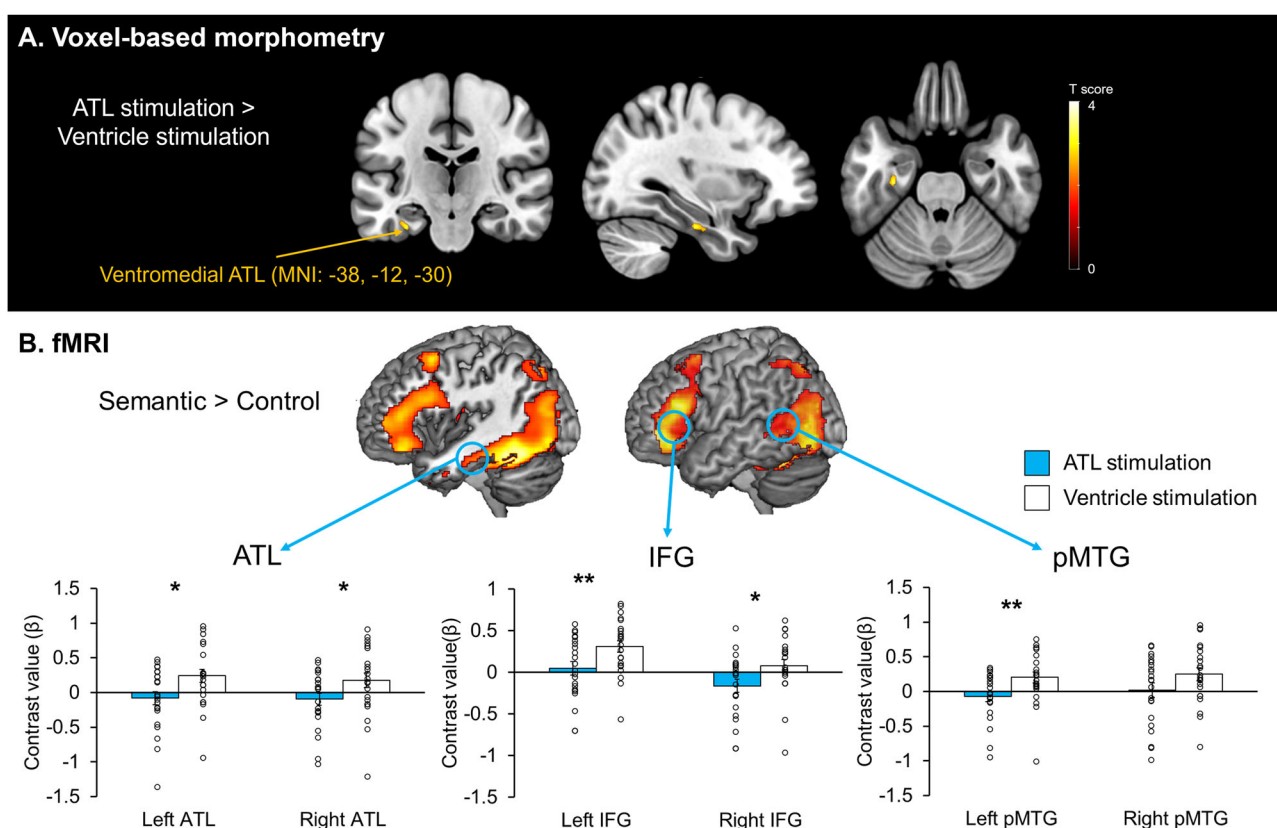

**Fig. 6 | Structural and functional MRI results. A** The results of VBM analysis. Colour bar represents t score. **B** Top: fMRI results of the contrast of interest (semantic > control) after ventricle stimulation. Bottom: Region of interest (ROI) analysis in the semantic network, including the ATL, IFG, and pMTG. Two-sided paired *t* tests showed significant changes in the ATL (left: *t*(21) = −2.52, *P* = 0.020, Cohen's *d* = −0.54, 95% Confidence Interval (CI) = [−0.587, −0.056]; right: *t*(21) = −2.18, *P* = 0.041, Cohen's *d* = −0.46, 95% CI = [−0.535, −0.012]), the IFG (left:

*t*(21) = −2.83, *P* = 0.010, Cohen's *d* = -0.60, 95% CI = [−0.454, −0.069]; right: *t*(21) = −2.69, *P* = 0.014, Cohen's *d* = -0.57, 95% CI = [−0.456, −0.054]), and the pMTG (left: *t*(21) = −2.68, *P* = 0.014, Cohen's *d* = -0.57, 95% CI = [−0.485, −0.061]; right: *t*(21) = −1.84, *P* = 0.079, Cohen's *d* = -0.39, 95% CI = [−0.489, 0.029]). Light blue bars represent the ATL stimulation. White bars indicate the control (ventricle) stimulation. Black circles represent individual data. Bars represent the mean and standard error of the mean for each condition. **\*\*P* < 0.01, \**P* < 0.05.

(Fig. S4). To examine the effects of ATL tbTUS, we performed region of interest (ROI) analysis using a priori ROIs in the semantic network, including the ATL, IFG, and pMTG. Planned paired *t* tests demonstrated that ATL tbTUS significantly decreased task-induced regional activation in the bilateral ATL (left: *t* = −2.52, *P* = 0.020; right: *t* = −2.18, *P* = 0.041), IFG (left: *t* = −2.83, *P* = 0.010; right: *t* = −2.69, *P* = 0.014), and pMTG (left: *t* = −2.68, *P* = 0.014, right: *t* = −1.84; *P* = 0.079) (Fig. 6B, Bottom). All data are summarised in the Source Data.

### Effects of tbTUS on effective connectivity within the semantic network

To investigate the effects of tbTUS within the semantic network, we employed dynamic causal modelling (DCM), a method that estimates and infers interactions among predefined brain regions across different experimental contexts[53]. DCM estimates three parameters: intrinsic connectivity (baseline connections), modulatory connectivity (task-related changes in connections), and driving inputs (direct effects of

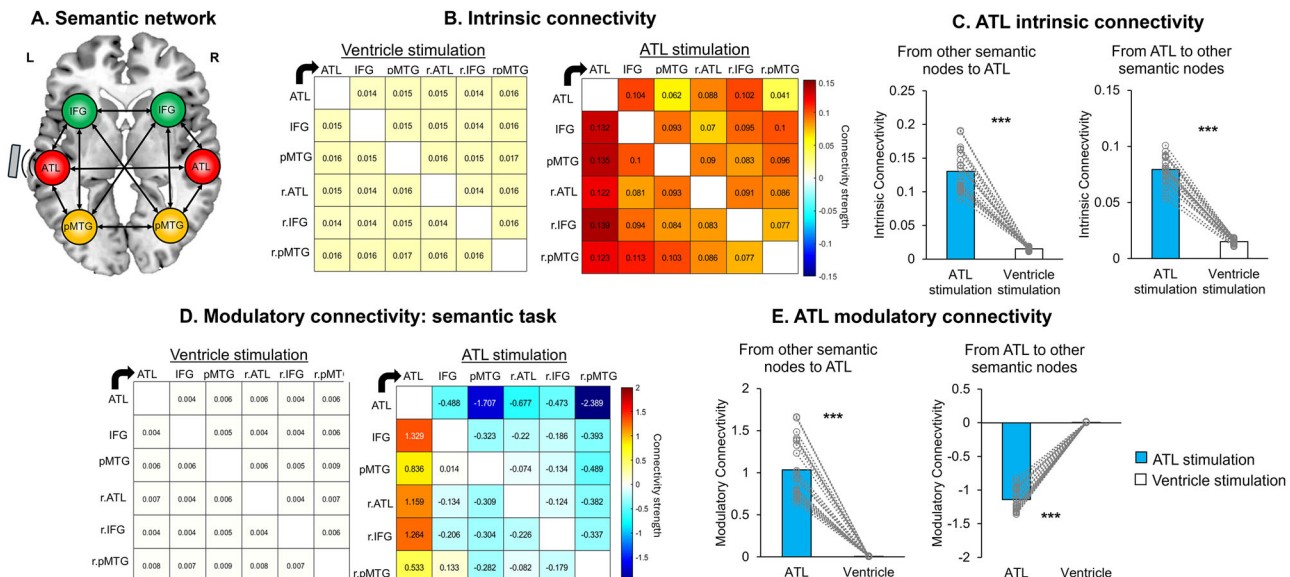

**Fig. 7 | DCM results. A** Model of connectivity within the semantic network. **B** Intrinsic connectivity results. tbTUS at the left ATL significantly increased the intrinsic connectivity of all connections within the semantic network compared to the control (ventricle) stimulation. Colour bars indicate the strength of effective connectivity among semantic nodes. **C** Averaged ATL intrinsic connectivity results. Two-sided $t$ tests showed significant increases in ATL intrinsic connectivity (from other nodes to ATL: $t(21) = 17.35$, $P < 0.001$, Cohen's $d = 4.00$, 95% Confidence Interval (CI) = [0.102, 0.130]; from ATL to other nodes: $t(21) = 27.95$, $P < 0.001$, Cohen's $d = 5.96$, 95% CI = [0.059, 0.069]). **D** Modulatory connectivity results during semantic processing. tbTUS at the left ATL significantly modulated modulatory connections compared to the control (ventricle) stimulation. Colour bars indicate the strength of effective connectivity among semantic nodes. **E** Averaged ATL modulatory connectivity results. Two-sided $t$ tests showed significant changes in ATL modulatory connectivity (from other nodes to ATL: $t(21) = 14.86$, $P < 0.001$, Cohen's $d = 3.10$, 95% CI = [0.886, 1.173]; from ATL to other nodes: $t(21) = -33.76$, $P < 0.001$, Cohen's $d = -7.04$, 95% CI = [-1.218, -1.077]). ATL tbTUS significantly increased the connection from other semantic regions to the ATL (target), whereas it decreased the connectivity from the ATL (target) to the other regions within the semantic network. Arrows represent the direction of connectivity. Blue bars indicate the ATL stimulation, and white bars represent the control (ventricle) stimulation. Bars represent the mean and standard error of the mean for each condition. ***$P < 0.001$.

external stimuli). The bilateral ATL, IFG and pMTG were defined as network nodes with mutual connections (Fig. 7A). Our model examined tbTUS effects at the left ATL in both baseline (intrinsic) and semantic task (modulatory) conditions, estimating 30 intrinsic parameters for baseline connectivity and 30 modulatory parameters representing connectivity changes due to tbTUS as the driving input.

Baseline intrinsic connections within the semantic network were significantly positive after ventricle stimulation, indicating mutual facilitation between semantic nodes (Fig. 7B). ATL stimulation significantly increased all intrinsic connectivity compared to the ventricle stimulation ($ts > 4.16$, ps $_{FDR-corrected} < 0.05$) (Fig. 7B and Table S2). To delineate the effects of ATL tbTUS on ATL connectivity, we averaged the connectivity from the ATL towards to other regions and from other regions to the ATL. ATL tbTUS significantly increased connectivity both from the ATL to other regions ($t = 27.95$, $P < 0.001$) and from other regions to the ATL ($t = 17.35$, $P < 0.001$) (Fig. 7C). All results are summarised in Supplementary Table S2 and Source Data.

Task-related modulatory connections were significantly positive after ventricle stimulation, reflecting mutual facilitation during semantic processing (Fig. 7D). ATL tbTUS increased connectivity from all other semantic nodes to the ATL (target) ($ts > 8.23$, ps $_{FDR-corrected} < 0.001$) but decreased most other connections ($ts < -2.57$, ps $_{FDR-corrected} < 0.05$), except for connections involving the left and right pMTG to the right ATL, left IFG, and their interconnections (Fig. 7D and Table S3). Averaged connectivity showed ATL tbTUS decreased connectivity from the ATL to other regions ($t = -33.79$, $P < 0.001$) but increased connectivity from other regions to the ATL ($t = 14.85$, $P < 0.001$) (Fig. 7C). The driving input, ATL tbTUS was also significant ($t = -71.78$, $P < 0.001$). All results are summarised in Supplementary Table S3 and Source Data.

## Aversive effects of tbTUS

No participants reported experiencing any discomfort during the experiment. Questionnaire analysis showed no significant differences between the ATL and ventricle stimulation ($\chi^2 < 4.326$, $Ps > 0.115$) (Table S4). Participants who experienced "unusual sensations on the skin" or "tingling" attributed these to the ultrasound gel, while those who reported "neck pain" linked it to the chair used during the session. Reports of "difficulty paying attention" and "sleepiness" were associated with the extended duration of the session. Only two participants reported "dizziness," which they attributed to the MRI scanning rather than the FUS. Approximately 78% of participants noted additional symptoms, but all described these as "hearing a faint clicking" or "tapping sound" during both ventricle and ATL stimulation sessions. No other symptoms related to tbTUS were reported, and participants were unable to differentiate between ATL and ventricular stimulations.

## Discussion

In this study, we provide convergent evidence that ATL-focused TUS can modulate the semantic system, leading to semantic memory enhancement. Our findings revealed structural, neurochemical and functional changes in the ventromedial ATL following tbTUS, which led to enhanced semantic memory performance. Importantly, these effects were specific to the ATL and semantic memory processing, while no comparable changes were observed following the control stimulation (ventricle), in the control region (OCC), or during control tasks involving visuospatial processing. Across multiple MRI modalities, we observed: increased grey matter volume in the ventromedial ATL, neurochemical alterations (e.g., decreased GABA and increased Glx, NAA, tCr, and choline), decreased task-induced regional activity, and enhanced facilitatory connectivity within the semantic network.

These findings support the proposal that tbTUS induces changes in cortical excitability, neurochemical balance, and functional connectivity, reflecting the long-term neuroplasticity potentially linked to LTP/LTD-like mechanisms[30,31,54]. Our study provides robust evidence for the potential of TUS as a therapeutic tool to treat memory impairments.

tbTUS targeting the ventromedial ATL resulted in a significant decrease in regional GABA concentrations while an increase in glx levels. These findings suggest that tbTUS modulates the excitation and inhibition balance (EIB) by regulating GABAergic and glutamatergic transmission within the target region. This aligns with evidence from animal studies showing that TUS regulates the expression of GABA and glutamate receptors, including NMDAR-mediated modulation of GABA receptor activity and BDNF signalling, which regulates both glutamatergic and GABAergic synapses[51,54]. These mechanisms are commonly associated with the induction of long-lasting synaptic plasticity, such as LTP and LTD, which are crucial for memory processing[55]. In previous human studies, a 40 s tbTUS of the PCC significantly decreased GABA concentrations[39] and TUS of the motor cortex altered the ratio of GABA and glx levels (EIB)[43]. Similarly, our previous investigation using inhibitory continuous theta burst stimulation (cTBS) over the ATL found increased GABA levels, which led to disruptions in semantic memory performance[24]. Together, our findings further support the roles of GABAergic inhibition and glutamatergic excitation in shaping semantic memory and its underlying neuroplasticity in the ventromedial ATL. The integrated excitatory-inhibitory mechanisms may be essential for refining semantic representation, leading to more precise and accurate semantic representation. By modulating the activity of both GABAergic interneurons and glutamatergic neurons, TUS can regulate overall excitation and inhibition in the region, promoting neuroplasticity in cognition functions[56–58].

MRS-measured GABA has been suggested to primarily reflects tonic inhibition rather than synaptic GABAergic signalling[59,60], though evidence remain inconsistent[61,62]. A recent study identified a link between MRS-detected GABA and phasic synaptic activity[63], suggesting that it may capture both tonic and synaptic GABAergic transmission. Similarly, MRS-Glx measures total glutamate and glutamine concentrations but cannot reliably distinguish between them. MRS-detectable glutamate is closely linked to energy metabolism, as it serves as a key metabolite in the tricarboxylic acid (TCA) cycle[64,65]. Neuronal activity and metabolism are highly interconnected, with glucose consumption and glutamate-glutamine cycling increasing in response to task-related activation[64,66]. Studies combining MRS and fMRI have shown that GABA and Glx concentrations across cortical regions are associated with stimulus-induced BOLD signal changes and functional connectivity (for a review, see ref. 67). Here, we demonstrated that tbTUS to the ATL decreased GABA and increased Glx, accompanied by reduced task-induced activity in the ATL and other semantic regions during semantic processing (Fig. 5B). We replicated previous work showing a relationship between GABA levels in the ATL, reduced task-induced BOLD signal changes, and improved semantic task performance[26]. Our finding that reduced task-induced ATL activity with better semantic performance supports the neural efficiency hypothesis, which suggests that more efficient cognitive processing is linked to reduced neural activation[68]. Together, our results indicate that ATL tbTUS enhances semantic memory by modulating regional GABA and glutamate levels, and reducing task-induced activity in the ATL, as well as in functionally connected regions such as the IFG and pMTG.

Our findings provide evidence that TUS can modulate neurometabolites such as NAA, tCr, and Cho, suggesting its potential to enhance neural function and cognition. NAA is a key marker of neuronal integrity, function and metabolism[69], with declines observed in neurodegenerative diseases like AD and dementia due to neuronal loss

and metabolic dysfunction[70–72]. The increase in NAA following tbTUS may indicate improved neuronal viability and neuroenergetic function. Similarly, the modulation of tCr suggests effects on brain energy metabolism, as Cr and phosphocreatine play a crucial role in adenosine triphosphate (ATP) production[73]. Given that Cr reductions have been linked to cognitive ageing and neurodegenerative conditions such as AD[74,75], an increase in tCr following tbTUS may indicate improved energy metabolism, potentially supporting synaptic plasticity and cognitive resilience. Changes in Cho levels further suggest potential effects on membrane turnover and neuroinflammation[60]. Elevated Cho has been associated with both gliosis and membrane degradation[76], and its modulation by tbTUS may reflect changes in phospholipid metabolism or cellular repair mechanisms[77]. Together, these findings suggest that tbTUS not only influences neurochemical balance but also contributes to energy metabolism and neuronal integrity. Further studies are needed to clarify the mechanisms underlying these neurometabolic changes and their long-term effects on brain function.

Effective connectivity analysis provides crucial directional information about interactions between brain regions, offering deeper insights into network dynamics[53]. Previous studies utilising a "perturb-and-measure" approach[78] have successfully shown that inhibitory rTMS can suppress neural activity at the stimulation site while enhancing task-related activity in the contralateral hemisphere and increasing connectivity from the non-stimulated hemisphere to stimulated hemisphere across motor[79,80], language[81,82], and semantic networks[52,83]. In this study, we applied facilitatory tbTUS to the left ATL and observed reduced task-related activity and enhanced intrinsic connectivity across the entire semantic network. Our findings suggest that tbTUS increased facilitatory drive from the left ATL to all other semantic regions and vice versa, strengthening network-wide interactions. During semantic processing, tbTUS also modulated modulatory connectivity, enhancing influence from other semantic regions to the left ATL while reducing connectivity from the left ATL to other regions. These results suggest that tbTUS-induced changes in modulatory connectivity reflect alterations in pre-existing intrinsic connectivity patterns. Notably, stimulation of the control site did not elicit significant task-related connectivity changes, reinforcing the idea that tbTUS directly modulates intrinsic connectivity, which in turn influences task-specific interregional interactions[52,84]. Providing key insights into the dynamic and rapid plasticity of intact neural networks, our findings suggest that TUS-induced fast adaptive reorganisation within the functional network may be essential for preserving and enhancing cognitive function. This mechanism is particularly relevant for understanding how tbTUS can facilitate long-term neural recovery following brain injury or neurosurgery by promoting adaptive plasticity within functional networks.

Our study demonstrated that a single session of tbTUS can induce structural alterations in GM, consistent with findings from animal studies showing TUS-induced structural changes[50,51,85]. In rodents, TUS has been shown to increase dendritic spine density in CA1 hippocampal neurons[51] and enhance memory function by increasing neuronal activity, promoting dendritic spine formation, and reducing spine elimination[50,85]. In this study, we found the increased GM in the ventromedial ATL following the ATL tbTUS, as detected by VBM. This finding is consistent with our previous work showing GM changes in the ATL after 40 seconds of cTBS[86]. VBM detects localised changes in GM and white matter (WM) volume, reflecting variations in voxel classification and potentially a combination of factors[87]. The underlying mechanisms of GM changes detected by VBM include axon sprouting, dendritic branching, synaptogenesis, neurogenesis, glial modifications, and angiogenesis[88]. Given the rapid time scale of the observed changes, they are more likely to reflect fast-adapting neuronal plasticity, such as modifications in synaptic morphology, dendritic integration, or interneuron density, rather than long-term structural remodelling. The local structural plasticity identified in this

study supports the notion that TUS can drive regional synaptic modifications in the adult human brain, providing further evidence of its potential to induce neuroplasticity.

There are several limitations in this study. First, our sample size is relatively small. Second, while we employed a tbTUS protocol that has frequently been associated with excitatory effects[39,41,89], accumulating evidence indicates that tbTUS can yield either facilitatory or inhibitory outcomes depending on brain state, and experimental context[44,46]. In this study, we demonstrated the facilitatory effects across behaviour, neurochemistry and fMRI measures, but further research is needed to validate and extend these findings. Third, MRS can detect the total concentration of GABA and Glx within a voxel of interest, but cannot differentiate between distinct pools of these neurochemicals[90]. In addition, the MRS-derived Glx signal reflects the combined tissue content of glutamate and glutamine, without distinguishing between the two[64]. Despite these limitations, linking GABA/Glx shifts with functional connectivity offers important insights into ultrasound-induced plasticity. Fourth, although ventricle-TUS served as an active control condition in this study and our previous work has validated its suitability by showing that it preserves the auditory and somatosensory features of active stimulation without altering brain structure or function[48], we acknowledge that additional controls (e.g., sham or off-target stimulation) would further strengthen the specificity of the conclusions. Finally, to validate our findings, future work should include longitudinal studies with long-term interventions to establish the persistence of ATL tbTUS effects.

Our findings provide fundamental insights into how TUS modulates neural networks, revealing its potential as a non-invasive tool for enhancing cognitive processing. By demonstrating structural, neurochemical and functional changes in the ventromedial ATL, we showed that tbTUS can regulate the excitation-inhibition interaction and facilitate adaptive network reorganisation. These results not only advance our understanding of the mechanisms underlying semantic memory but also have significant implications for the development of neuromodulation therapies to enhance semantic memory. Given its ability to selectively modulate neurochemical systems and functional connectivity, TUS holds promise as a novel intervention for cognitive enhancement and neurorehabilitation in clinical populations, including individuals with dementia and other neurodegenerative disorders.

## Methods

All research procedures complied with relevant ethical regulations. Informed written consent was obtained from all participants prior to the study, which was approved by the Ethics Committee of the School of Psychology at the University of Nottingham (F1417).

### Participants

Twenty-three healthy young adults (6 males, mean age = 21 ± 3.02 years, ranging from 19 to 33) were recruited for this study. All participants were right-handed English speakers[91] with no current diagnosis of neurological or psychiatric disorders, and were not taking any medications known to influence brain chemistry during the study. The sample size was determined based on a previous study[39], which indicates that for a significant level of α = 0.05, power = 80%, at least 18 participants were required to detect tbTUS effects compared to sham. One participant completed only first and second sessions (ATL stimulation). Participants were compensated £30 for completing each session.

### Experimental design and procedure

Figure 1 summarises the study design and procedures. Participants were asked to attend three sessions at the same time of the day, at least 5 days apart, to prevent carry-over effects of tbTUS. The first session was to acquire a high-resolution T1-weighted MR image for neuronavigation, while the subsequent sessions applied tbTUS targeting either the left ventromedial ATL or ventricle as a control site. The order of stimulation was counterbalanced across participants (Fig. 1A). Participants were informed that stimulation would be applied to different areas of the semantic network, although experimenters were aware of the stimulation site to ensure accurate targeting. To check blinding integrity, participants were verbally asked at the end of all sessions, after being informed that one session involved control stimulation, whether they could distinguish between the two stimulation conditions. No participants were able to identify the control session.

In active tbTUS sessions, participants performed a semantic association task and pattern matching as a control task prior to tbTUS. These tasks have been extensively validated in both healthy and dementia populations[92–94] and previously utilised to investigate the relationship between semantic task performance, neurochemical levels, and neural activation[24–26] (Fig. 1B). The semantic association decision task required participants to determine which of two pictures displayed at the bottom of the screen was more closely related in meaning to a probe picture shown at the top. For the pattern-matching task, scrambled versions of the pictures used in the semantic association task were created. In this task, participants identified which of two patterns at the bottom matched a probe pattern presented at the top. Participants responded by pressing one of two buttons corresponding to their choice. Each trial began with a 500 ms fixation period, followed by a 4500 ms presentation of the stimuli. Each task consisted of 116 trials. Participants were instructed to respond as quickly and accurately as possible by pressing a key. Stimuli were not repeated across the pre- and post-stimulation sessions. Immediately after tbTUS, participants went to the MRI centre next to the lab and had a multimodal imaging.

For MRI, we employed the same multimodal imaging paradigm as in our previous studies[24,26]. The imaging protocol included structural, biochemical (MRS), and functional imaging. During structural imaging and MRS acquisition, participants were instructed to remain relaxed with their eyes open. For fMRI, participants completed the semantic and control tasks in a block design. Each task block consisted of four trials, and eleven blocks of each task were alternated (e.g., A-B-A-B) with a 4000 ms fixation period between blocks. Each trial began with a 500 ms fixation period, followed by a 3500 ms presentation of the stimuli. Each task had 44 trials, and the total fMRI task took about 7 min 30 s. After the scanning session, participants returned to the lab to repeat the tasks. Task presentation and response recording were implemented using PsychoPy (version 2024.2.1). Finally, aversive response questionnaires related to TUS were administered at the end of the study.

### Transcranial focused ultrasound stimulation

We used the four-element CTX-500-4CH transducer coupled with the NeuroFus PRO TPO-203 system (Sonic Concepts, Brainbox Ltd., Cardiff, United Kingdom; fundamental frequency: 0.5 MHz; diameter: 64 mm). TUS was delivered using identical theta-burst parameters as follows: central frequency = 500 kHz, pulse duration = 20 ms, pulse repetition interval = 200 ms, pulse repetition frequency = 5 Hz, duty cycle = 10%, ISPPA = 54.51 W/cm2, total duration = 80 sec, steering depth = variable (ATL = 5.31 ± 0.72 cm; ventricle = 5.28 ± 0.34 cm) (Fig. 2A).

TUS was delivered to either the ATL or the ventricle in the left hemisphere. The ATL target was determined based on coordinates (MNI: −36, −15, −30) derived from prior TMS-fMRI studies[24,52]. The ventricle target was manually identified at the centre of the left lateral ventricle using each participant's T1-weighted MRI image, as achieved in previous work[48]. Transducer positioning over the head was determined for each participant by using a novel and validated open-source tool, which identifies the lower head-transducer angle positioning[95] (https://github.com/CyrilAtkinson/TUS_entry). TUS preparation involved filling the transducer with degassed water and then applying ultrasound transmission gel to both the head and the transducer.

When required, air bubbles in the gel were manually removed by using a syringe.

To verify accurate targeting, acoustic simulations were conducted for all participants using k-plan V1.2 (Brainbox Ltd., Cardiff, United Kingdom). The stimulations used both the acquired T1-weighted and a pseudo-computed tomography (PCT) scan that was estimated for each participant from the T1-weighted scan[96]. Simulations confirmed precise targeting of both the ATL and ventricle in all participants (Fig. 2B). Group-level results indicated a peak pressure at the target of $578.22 \pm 101.27$ kPa for the ATL stimulation and $624.6 \pm 90.0$ kPa for the ventricle stimulation. The ISPPA in situ was $11.46 \pm 4.09$ W/cm² for the ATL stimulation and $13.26 \pm 4.19$ W/cm² for the ventricle stimulation. The transcranial mechanical index was $0.80 \pm 0.16$ (max = 1.11) for the ATL stimulation and $0.88 \pm 0.13$ (max = 1.33) for the ventricle stimulation. The maximal temperature increase was well within established safety guidelines (ATL at target = $0.17 \pm 0.09$, max=0.47; ATL at tissue = $1.95 \pm 0.43$, max = 2.88; ATL at skull = $3.07 \pm 0.73$, max = 4.69; ventricle at target = $0.27 \pm 0.08$, max = 0.567; ventricle at tissue = $0.63 \pm 0.07$, max = 0.761; ventricle at skull = $1.05 \pm 0.13$, max = 1.33)[97].

## Magnetic resonance imaging acquisition

Images were acquired using a General Electric (GE) SIGNA Premier 3 T MR scanner with a 48-channel head coil (GE Healthcare, USA). T1-weighted images were obtained using 3D MPRAGE sequence (voxel size = 1 mm isotropic, field of view [FOV] = 256, matrix = 256, 256 sagittal slices, inverse time [TI] = 800 ms, flip angle [FA] = 8°).

MRS data were collected using a GABA-edited MEGA-PRESS sequence (TR = 2000 ms, TE = 68 ms)[98]. The voxel of interest (VOI) was manually positioned in the left ventrolateral ATL (voxel size = $40 \times 20 \times 20$ mm), avoiding overlap with the hippocampus and occipital cortex (OCC) (voxel size = $30 \times 30 \times 30$ mm) (Fig. 1A). Spectra were acquired in interleaved blocks of four scans with the MEGA inversion pulses applied at 1.95 ppm to edit the GABA signal. This included 184 repeats at the ATL VOI and 112 repeats at the OCC VOI. The protocol provided robust measurements of GABA and glx concentrations from the ATL VOI[26,99,100]. A total of 4096 sample points were collected at a spectral width of 5 kHz.

fMRI data were collected using a whole- brain 2D GE-EPI sequence (TR = 1400 ms, TE = 35 ms, flip angle = 68°, in-plane FOV = $212 \times 212$ mm, 57 slices, slice thickness = 2 mm, voxel size = 2 mm isotropic, hyperband factor = 3, ARC factor = 2, 344 volumes, 8.2 mins total scan time). To account for echo-planar imaging (EPI) distortions, two SE-EPI images with opposite phase encoding directions will be acquired, sharing the same geometry, echo spacing, and phase encoding direction parameters as the GE fMRI scans.

## VBM analysis

Voxel-based morphometry (VBM) was utilised to investigate changes in grey matter (GM) and white matter (WM) using CAT12. Preprocessing steps included segmentation, normalisation, modulation and smoothing[101] (Fig. 1D). Data were acquired from two timepoints (ATL and ventricle stimulation sessions) for each subject. To reduce intra-subject variability and improve registration accuracy, an unbiased within-subject average image was generated from both timepoints. Subsequent segmentation and normalisation were performed on this average image, ensuring longitudinal consistency in tissue classification. The normalisation process used an optimised DARTEL-based approach, estimating high-dimensional warping parameters from the average image and applying them to each individual scan. This approach improved alignment of individual GM and WM maps to a customised template in standard MNI space[102]. Images were corrected for bias field inhomogeneities and segmented into GM, WM, cerebrospinal fluid (CSF), and non-brain tissues. To account for local volume changes, Jacobian modulation was applied, preserving the absolute GM and WM volume at each voxel[87]. All images were smoothed with a 5 mm full-width at half maximum (FWHM) isotropic Gaussian kernel.

Voxel-wise statistical analysis was conducted using the general linear model (GLM) to examine specific regional GM and WM differences between ATL and ventricle stimulation. Paired $t$ tests were performed on each participant's scans, comparing tbTUS sites (ATL vs. ventricle) while controlling for total intracranial volume. We applied a statistical threshold of $P < 0.005$ (uncorrected) at a voxel level. Based on prior findings[86], we hypothesised that tbTUS targeting the left ATL would lead to morphological changes in this region. An a priori region of interest (ROI) was defined as a 5 mm sphere in the left ventral ATL (MNI: $-36$, $-15$, $-30$). Multiple comparisons were corrected using a family-wise error (FWE) threshold of $p < 0.05$ after small volume correction (SVC) within the defined region.

## MRS analysis

MRS spectra were analysed using GANNET 3.3.1[103]. Spectra underwent frequency and phase correction via spectral registration[104] and were filtered with 3-Hz exponential line broadening (Fig. 1C). The area under the edited GABA peak at 3 ppm, referred to as GABA+ (GABA + macromolecules), was estimated because the edited GABA signal is contaminated by co-edited macromolecules. GABA + , glx, and unsuppressed water signals were modelled using a single Gaussian peak with a 5-parameter Gaussian model and a Gaussian-Lorentzian model, respectively, and integrated. Additional neurochemicals, including NAA (N-acetylaspartate), Cho (choline), and tCr (total creatine), were also quantified. MRS voxels were coregistered to T1-weighted images, and segmentations into GM, WM, and CSF were used to compute tissue-corrected neurochemical concentrations. The correction assumed negligible neurochemical levels in CSF and twice the concentration in GM compared to WM[105]. Neurochemical concentrations were normalised to water and adjusted using the ratio GM/(GM + WM). Data quality was assessed for each voxel using metrics such as fit errors, signal-to-noise ratio (SNR), and linewidth. Data exclusion criteria included fit errors > 15% for each neurochemical, water linewidths (FWHM) > 20 Hz, and SNR < 40. For the ATL VOI, 7 spectra were excluded (3 from ATL tbTUS sessions and 4 from ventricle tbTUS sessions). For the OCC VOI, 8 spectra were excluded (6 from ATL tbTUS sessions and 2 from ventricle tbTUS sessions). Table S1 provides a summary of the data quality metrics for the spectra retained after visual inspection and tissue segmentation analysis.

## fMRI analysis

fMRI data were pre-processed using the latest version of the NIHR Nottingham BRC imaging pipeline, as detailed on their GitHub page (https://github.com/SPMIC-UoN)[106]. This pipeline incorporates and utilises tools from leading advanced image analysis software, including FSL, ANTs, SPM, and FreeSurfer. Preprocessing steps included EPI distortion correction, motion correction, slice timing adjustment, segmentation, normalisation, and smoothing with a 5 mm FWHM Gaussian kernel.

Statistical analysis was conducted using GLM in SPM12 (https://www.fil.ion.ucl.ac.uk/spm/software/spm12/). The design matrix included task conditions (semantic and control) and six motion parameters as regressors. A contrast of interest (semantic > control) for each participant was calculated. A one-sample $t$ test was performed to estimate the contrast of interest at the group level. A standard voxel-wise analysis with cluster-level inference was performed, applying a voxel-wise threshold of $P < 0.001$, corresponding to T > 5. Clusters were considered significant when passing a threshold of $P$ FWE-corrected <0.05, with at least 100 contiguous voxels.

Region of interest (ROI) analysis was carried out using Marsbar[107], including the ATL, IFG, and pMTG. Peak coordinates were based on prior studies using the same tasks: ATL (MNI left: $-36$, $-15$, $-30$; right:

33, −6, −36), IFG (MNI left: −48, 21, 24; right: 57, 24, 21), and pMTG (MNI left: −57, −48, −3; right: 54, −69, 12).[24–26] Planned paired $t$ tests were conducted to examine tbTUS effects between ATL and ventricle stimulation sessions.

### Dynamic causal modelling (DCM)
DCM analysis was conducted using SPM12 to assess the effects of tbTUS on effective connectivity within the semantic network. A single model with six nodes (bilateral ATL, IFG, and pMTG) was tested, as ATL tbTUS influenced both the target region and functionally connected semantic areas. For each participant and target node, blood oxygenation level-dependent (BOLD) time series were extracted and converted to neural activity using the first eigenvector in SPM12.

The intrinsic model included bidirectional connections among all six nodes (Fig. 6A), testing 22 parameters representing baseline connectivity without experimental perturbations. The modulatory model examined task-related changes in connectivity, with a driving input at the left ATL (e.g., ATL tbTUS), testing 22 modulatory parameters. Analyses were conducted separately for ATL and ventricle stimulation during the semantic and control tasks, focusing on the effects of ATL stimulation. We then extracted individual-specific estimates for model parameters, including the driving input and the strength of intrinsic and modulatory connectivity, to evaluate the effects of tbTUS at the left ATL on both connectivity. These parameters were entered into FDR-corrected one-sample $t$ tests to assess differences from zero and paired $t$-tests to assess the ATL tbTUS on interregional connectivity.

### Statistical analysis
The statistical analyses were performed using IBM SPSS Statistics for Windows, version 28 (IBM Corporation, Armonk, NY, USA), R and SPM12. The questionnaire to evaluate the aversive effects of TUS was analysed using $\chi^2$ test.

For behavioural data, accuracy and reaction time (RT) were calculated for each individual. A $2 \times 2$ repeated measures ANOVA with stimulation (ATL vs. ventricle) and session (PRE vs. POST) as within-subject factors was performed on each task (semantic and control). Planned paired $t$ tests were conducted to examine tbTUS effects in pre- and post-stimulation sessions.

For MRS data, we performed planned paired $t$ tests (one-tailed) for GABA+ and glx following our hypothesis. For other neurochemicals such as NAA, tCr, cho, paired $t$ tests (two-tailed) were conducted.

### Reporting summary
Further information on research design is available in the Nature Portfolio Reporting Summary linked to this article.

## Data availability
The data generated in this study have been deposited in the Open Science Framework database under the CC BY 4.0 license (https://doi.org/10.17605/OSF.IO/FVK7C)[108]. Source data are provided with this paper.

## Code availability
The open-source tool used to determine transducer positioning is available at (https://github.com/CyrilAtkinson/TUS_entry)[95].

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

## Acknowledgements
The authors acknowledge Sarah Wilson, Louise Cowell, Andrew Cooper, Jan A Paul, and Mehri Kaviani for their MRI support in the project. J.J. was supported by the AMS Springboard (SBF007\100077). M.K. and C.A. were supported by the Engineering and Physical Sciences Research Council (EP/W004488/1, EP/X01925X/1 and EP/W035057/1). M.K. was also supported by the Guangci Professorship Program of Rui Jin Hospital (Shanghai Jiao Tong University). J.J., M.A.L.R., and M.K. were supported by the EPSRC & MRC-funded NEUROMOD + . M.A.L.R. is supported by MRC intramural funding (MC_UU_00030/9).

## Author contributions
Conceptualisation: J.J. and M.A.L.R.; methodology: J.J., C.A., and M.K.; investigation: J.J. and C.A.; writing—J.J. and C.A.; writing—review and editing: J.J., C.A., M.K., and M.A.L.R.

## Competing interests
The authors declare no competing interests.
