## [Transparent Peer Review file · Nature Communications]

Transcranial focused ultrasound stimulation enhances semantic memory by modulating brain morphology, neurochemistry and neural dynamics

Corresponding Author: Dr JeYoung Jung

Version 0:

Reviewer comments:

Reviewer #1

(Remarks to the Author)

This study has the potential to present very compelling evidence that transcranial focused ultrasound stimulation (TUS) targeting the left ventromedial anterior temporal lobe (ATL) can enhance semantic memory in healthy adults. The authors employ a multimodal approach; combining MRS, fMRI, and VBM; to demonstrate that a 5Hz patterned TUS protocol modulates neurochemical markers (e.g., reduced GABA, increased Glx, NAA, choline, etc), alters brain structure, and affects functional connectivity within the semantic network. These findings could support TUS as a promising NIBS technique with potential therapeutic applications and as such the paper is novel, timely and interesting.

However, at present, the paper lacks sufficient detail for a robust assessment of the scientific validity and reproducibility of the results. In addition, several analytical choices limit the interpretability of the findings, particularly in the ROI-based analyses and statistics. Below are major and minor points for the authors' consideration:

Major comments

1. Data availability and reproducibility

The manuscript does not provide any raw or preprocessed data, making it difficult to assess the robustness and reproducibility of the findings. At minimum, raw MRS data, fMRI and behavioral data, and analysis code (e.g., for behavioral, DCM, ROI, and MRS processing) should be made openly available. I strongly recommend depositing these in public repositories (e.g., OpenNeuro, Zenodo, GitHub) in line with Nature Communications data transparency standards.

2. Methodological transparency

The Methods section is significantly underdeveloped. Critical experimental details are missing:

- Please provide a comprehensive explanation of the TUS intervention, neuronavigation, including coupling methods with the head and the variability in acoustic parameters (e.g., average \pm SD of steering depth).
- The description of the ultrasound setup is vague (e.g., "manually removing air bubbles" in the gel is unclear). Please clarify.
- Include thermal rise estimates specifically for the skull, which is essential for safety evaluation. The max value has to be reported across the entire head.
- It is not clear what Mechanical Index is here. Is it PNP derated by 0.3dB/cm/MHz and divided by the $\sqrt{f_0}$? Because this is the definition of mechanical index. Is it the PNP derated by some other derating factor for the skull and divided by the $\sqrt{f_0}$? This is then MI_{tc}. If it is MI_{tc}, then it should be labelled MI_{tc}.
- Report where ISPPA and MI values are computed (target MNI coordinate vs. ROI peak).

2. Task design and execution

- The semantic task parameters inside the scanner (duration, jittering strategy, timing of stimuli) should be fully described in both the Methods and figure legends.
- Please clarify if the task design was optimized using established jittering procedures (e.g., optseq). It doesn't look like it is but that would be problematic for event related design unless it's a block design.
- Indicate total task time and block structure in the figures for clarity.
- Indicate how VBM was performed in the figure too.

3. fMRI and ROI analyses

- fMRI preprocessing and statistical thresholds are underreported. What Z or T threshold was used before correction?
- The use of spherical peak-based ROIs is problematic in the context of highly focal stimulation. I suggest basing ROI selection on the acoustic field with a minimum pressure threshold of 150 or 200 kPa.

4. MRS and statistical interpretation

All MRS results need correction for multiple comparisons.

Please provide spectra for all excluded MRS datasets in the supplement.

Include a collinearity matrix for all MRS measures and explore relationships between MRS metrics and behavior (e.g., correlations or regression models).

Report behavioral correlations with MRS ratios (e.g., GABA/Glx) and ISPPA/pressure at the MRS voxel.

Consider whether GLMs accounting for session order effects (e.g., ATL first vs. ventricle first) would be more appropriate than t-tests.

5. Excitatory/inhibitory framing

The framing of tbTUS as “excitatory” needs refinement. There are examples of the same 5 Hz protocol yielding inhibitory responses depending on context and state. A more balanced introduction and discussion, acknowledging variability in outcomes, is essential.

6. The Introduction should include a clear justification for targeting the left ATL specifically. While the ATL's role in semantic memory is well supported, its bilateral involvement is well-documented, and the rationale for left-lateralized stimulation is missing. This omission limits the reader's ability to interpret whether the observed effects are due to lateralized processing or methodological constraints. It should be made clear what could happen with bilateral stimulation.

7. Results: The DCM and VBM analyses should be re-evaluated with ROIs grounded in the simulated acoustic field rather than anatomical peaks.

Minor comments

- Clarify whether the study was single- or double-blinded for both participants and experimenters. I assume neither, but this should be clear upfront (in the abstract)
- The claim: “To our knowledge, this is the first study to demonstrate effective connectivity changes...” is inaccurate and should be removed or revised.
- The term “theta-burst transcranial ultrasound stimulation (tbTUS)” is misleading. As ITRUSST discourages the term “burst” for pulse trains, consider using “5Hz patterned stimulation” instead.
- Clarify whether group-level acoustic parameters (e.g., 191.9 ± 28.06 kPa, ISPPA 1.25 ± 0.4 W/cm²) are computed ROI-wise or are derived from peak locations. As currently stated, they lack interpretability. The mean intensity and MItc are not very meaningful at the group level, unless the authors can make a claim about these. I would remove and instead focus on interindividual variability
- The legend of Figure 2 incorrectly describes acoustic values; please double-check the stated pressure ranges.
- Improve the figure labelling to include task design details, stimulus timing, and data collection timelines.
- The VBM section has a grammar issue: “A statistical threshold of $p < 0.005$ (uncorrected).” Is not a sentence.

Reviewer #2

(Remarks to the Author)

This study presents evidence of the neuromodulatory effects of theta-burst transcranial ultrasound stimulation (tbTUS) on semantic memory, utilizing multimodal neuroimaging to reveal structural, biochemical, and functional changes. While the findings are promising, several key issues need further clarification to enhance the manuscript's impact and ensure its reproducibility.

Major comments

1. The study enrolled 23 participants, but the final analyses included only 21 (after exclusions). Given the complexity of multimodal imaging and the subtlety of neurochemical/functional changes, the sample size raises concerns about statistical power, particularly for subgroup analyses (e.g., MRS exclusions). Provide a formal power analysis to justify the sample size, especially for detecting interactions (e.g., stimulation \times session effects).
2. Multiple researches have reported that different parameter settings of TUS can lead to varying neurofunctional regulatory effects (excitatory or inhibitor). More information is needed on how the tbTUS stimulation parameters were chosen in this study and whether they were optimized for the target of ATL and the specific cognitive task. The relationship between parameters and effects should be mentioned in the discussion.
3. Ventricle stimulation was used as a control site. While prior work supports this approach (Atkinson-Clement et al., 2024), the manuscript does not fully address whether ventricular stimulation itself may influence cerebrospinal fluid dynamics or adjacent neural structures (e.g., thalamocortical pathways). Include additional controls (e.g., sham stimulation or off-target stimulation) to rule out nonspecific effects of ultrasound energy propagation will enhance the credibility of the conclusions.
4. MRS findings (e.g., decreased GABA+/increased Glx) are interpreted as shifts in excitation-inhibition balance. However, MRS cannot distinguish synaptic vs. extrasynaptic GABA/glutamate pools or differentiate glutamine from glutamate contributions. I suggest the authors discuss these limitations. Linking neurochemical shifts (GABA/Glx) to functional connectivity changes will advance the understanding of ultrasound-induced plasticity.
5. The reported grey matter (GM) volume increases in the ATL after a single tbTUS session are striking. VBM is sensitive to hydration, blood flow, and registration errors. The manuscript does not rule out transient hemodynamic or fluid shifts as contributors. Include longitudinal follow-up scans to assess the persistence of GM changes and correlate them with behavioral outcomes.
6. The manuscript mentions the potential role of tbTUS in enhancing cognitive function, but the long-term effects of this intervention are not adequately discussed.

Minor comments

1. Specify the criteria for MRS data exclusion (e.g., linewidth thresholds) in the main text, not just supplementary materials.
2. The semantic task lacks difficulty titration. Variability in baseline performance (Fig. S1) suggests floor effects.
3. Consider adding a schematic summarizing the proposed mechanism (Fig. 6 is dense).

Reviewer #3

(Remarks to the Author)

This is an impressive set of results from a complex experiment that investigates effects of tbTUS on semantic processing networks across behaviour and multiple imaging modalities. The manuscript is well written. The authors demonstrate that tbTUS to the left anterior temporal lobe has a positive effect on semantic processing task performance, alongside increasing grey matter volume, modulating neurochemical markers and altering task-related BOLD signal across semantic brain networks.

Major

Behavioural task

More detail is needed about the behavioural tasks. For example: How many trials/blocks were there? Was the paradigm for the PRE and POST sessions the same as the in-scanner session (e.g. number of trials)? Were the stimuli the same or different for each session? Were participants instructed to respond as rapidly as possible?

If the stimuli used were the same each time the task is performed, is it possible that stimulation was acting through some mechanism other than directly improving semantic associative processing, but instead, for example, episodic learning over sessions?

An increase with TUS of ~4% accuracy is only about 1 trial better accuracy. How meaningful/robust is this finding? In Supplementary figure 1, it appears that there is one participant who improves much more than others. Is the result still significant if this participant is removed?

VBM

how was the SVC (MNI: -36, -15, -30) chosen? How does it relate to i) the stimulated region and ii) known semantic networks from prior activation (e.g. fMRI/icEEG) studies?

fMRI: What was task performance like in the scanner?

Since the duration of TUS effects on behaviour and brain remains poorly understood, it would be helpful to know if there was any effect of stimulation order on the results?

There is no discussion about whether the multiple behavioural and imaging effects of tbTUS were related to each other across participants. The questions are obvious (e.g. did changes in effective connectivity relate to behaviour/MRS/VBM effects?), although there are likely to be power issues. Nevertheless, some mention of how the behavioural and imaging results relate to each other seems important.

The Discussion should include a section exploring potential weaknesses of the study.

Minor

line 118: Typo "Combined with MRS enables..."

line 158: "Left ventricle" should presumably read "Left lateral ventricle"

Results section: In the first paragraph it would help to clarify that scanning took place in both session 2 and session 3 and this should also be added to Fig 1a.

Figure 1a) This could be clearer in general. For example, show that patients were randomised into either the ventricle then ATL group or the ATL then ventricle group.

Behavioural results: The ANOVA should really include factors of stimulation, session AND task.

Figure 4: The caption says that error bars represent standard error. There are no error bars on the figure.

Figure 6e: The white bars are missing.

Version 1:

Reviewer comments:

Reviewer #1

(Remarks to the Author)

I have now reviewed this manuscript for the second time. The authors have addressed many of my previous comments, and the revisions have substantially improved the clarity and coherence of the paper. The study presents interesting and potentially important findings regarding the effects of TUS on semantic memory, supported by a comprehensive multimodal approach.

However, there remain a few issues that should be addressed before I can endorse the manuscript for publication:

Data availability and documentation

The online dataset is available, but there is no accompanying metadata, readme file, or explanation of the task structure, data organization, or variable definitions. There are also no acquisition parameter summaries or analysis instructions. These materials are essential for reproducibility and transparency. The dataset must be improved before publication to ensure it meets FAIR principles and open-research expectations for findability, accessibility, interoperability, and reusability.

Task validation

Please add a brief statement acknowledging that the semantic association task has been extensively validated in previous studies by the authors' group, but that its generalizability to independent implementations by other laboratories remains to be established. This clarification will strengthen transparency and situate the work appropriately within the broader literature.

Blinding procedures

The manuscript states that the study was single blind, but it remains unclear how blinding integrity was verified. Please explicitly describe how blinding was checked. If this was not formally assessed, this limitation should be acknowledged in the discussion.

Reviewer #2

(Remarks to the Author)

The results of this study provide fundamental evidence that TUS is a valuable non-invasive tool for regulating cognitive processes by selectively modulating neurochemical systems and functional connectivity. In the revised manuscript, the authors have refined the experimental design and enhanced the clarity of the descriptions, added more statistical analyses and detailed discussions of the results. I think the current revisions has met the standards for publication.

Reviewer #3

(Remarks to the Author)

The authors have satisfactorily addressed all my comments.

Reviewer #1 (Remarks to the Author):

This study has the potential to present very compelling evidence that transcranial focused ultrasound stimulation (TUS) targeting the left ventromedial anterior temporal lobe (ATL) can enhance semantic memory in healthy adults. The authors employ a multimodal approach; combining MRS, fMRI, and VBM; to demonstrate that a 5Hz patterned TUS protocol modulates neurochemical markers (e.g., reduced GABA, increased Glx, NAA, choline, etc), alters brain structure, and affects functional connectivity within the semantic network. These findings could support TUS as a promising NIBS technique with potential therapeutic applications and as such the paper is novel, timely and interesting.

Response: We appreciate R1's positive evaluation of our manuscript.

However, at present, the paper lacks sufficient detail for a robust assessment of the scientific validity and reproducibility of the results. In addition, several analytical choices limit the interpretability of the findings, particularly in the ROI-based analyses and statistics. Below are major and minor points for the authors' consideration:

Major comments

1. Data availability and reproducibility

The manuscript does not provide any raw or preprocessed data, making it difficult to assess the robustness and reproducibility of the findings. At minimum, raw MRS data, fMRI and behavioral data, and analysis code (e.g., for behavioral, DCM, ROI, and MRS processing) should be made openly available. I strongly recommend depositing these in public repositories (e.g., OpenNeuro, Zenodo, GitHub) in line with Nature Communications data transparency standards.

Response: We appreciate R1's point. We have deposited all raw data and analysis code on OSF.

Accordingly, we have revised Data and materials availability, p18

“Data and analysis code are available on OSF (<https://doi.org/10.17605/OSF.IO/FVK7C>).”

2. Methodological transparency

The Methods section is significantly underdeveloped. Critical experimental details are missing:

- Please provide a comprehensive explanation of the TUS intervention, neuronavigation, including coupling methods with the head and the variability in acoustic parameters (e.g., average \pm SD of steering depth).

Response: We apologize for the lack of clarity in our original description. We have added more detailed explanation for TUS section.

We have revised Method and Materials.

Materials and Methods, Transcranial focused ultrasound stimulation, p20

“We used the four elements CTX-500-4CH transducer coupled with the NeuroFus PRO TPO-203 system (Sonic Concepts, Brainbox Ltd., Cardiff, United Kingdom; fundamental frequency: 0.5MHz; diameter: 64mm). TUS was delivered using an identical 5 Hz-patterned, theta-burst protocol for each participant, as follows: central frequency = 500 kHz, pulse duration = 20 ms, pulse repetition interval = 200 ms, pulse repetition frequency = 5 Hz, duty cycle = 10%, ISPPA = 54.51 W/cm², total duration = 80sec, steering depth = variable (ATL = 5.31±0.72cm; ventricle = 5.28±0.34cm) (Fig. 2A).

TUS was delivered to either the ATL or the ventricle in the left hemisphere. The ATL target was determined based on coordinates (MNI: -36, -15, -30) derived from prior TMS-fMRI studies^{24,48}. The ventricle target was manually identified at the centre of the left lateral ventricle using each participant’s T1-weighted MRI image, as achieved in previous work⁴⁵. Transducer positioning over the head was determined for each participant by using a novel and validated open-source tool which identifies the lower head-transducer angle positioning⁸⁶ (https://github.com/CyriAtkinson/TUS_entry). Neuronavigation was performed using Visor2™ V2.7 (Ant-Neuro, Hengelo, Netherlands). TUS preparation involved filling the transducer with degassed water and then applying ultrasound transmission gel to both the head and the transducer. When required, air bubbles in the gel were manually removed by using a syringe.

To verify accurate targeting, acoustic simulations were conducted for all participants using k-plan V1.2 (Brainbox Ltd., Cardiff, United Kingdom). The simulations used both the acquired T1-weighted and a pseudo computed tomography (PCT) scan that was estimated for each participant from the T1-weighted scan⁸⁷. Simulations confirmed precise targeting of both the ATL and ventricle in all participants (Fig. 2B). Group-level results indicated a peak pressure at target of 578.22 ± 101.27 kPa for the ATL stimulation and 624.6 ± 90.0 kPa for the ventricle stimulation. The ISPPA in situ was 11.46 ± 4.09 W/cm² for the ATL stimulation and 13.26 ± 4.19 W/cm² for the ventricle stimulation. The transcranial mechanical index was 0.80 ± 0.16 (max = 1.11) for the ATL stimulation and 0.88 ± 0.13 (max = 1.33) for the ventricle stimulation. The maximal temperature increase were well within established safety guidelines (ATL at target = 0.17 ± 0.09, max=0.47; ATL at tissue = 1.95 ± 0.43, max=2.88; ATL at skull = 3.07 ± 0.73, max=4.69; ventricle at target = 0.27 ± 0.08, max=0.567; ventricle at tissue = 0.63 ± 0.07, max=0.761; ventricle at skull = 1.05 ± 0.13, max=1.33)⁸⁸.”

• The description of the ultrasound setup is vague (e.g., "manually removing air bubbles" in the gel is unclear). Please clarify.

Response: Please, see the reply above.

- Include thermal rise estimates specifically for the skull, which is essential for safety evaluation. The max value has to be reported across the entire head.

Response: We have added the temperature rise estimation. Please, see the reply above.

- It is not clear what Mechanical Index is here. It is PNP derated by 0.3dB/cm/MHz and divided by the $\sqrt{f_0}$? Because this is the definition of mechanical index. Is it the PNP derated by some other derating factor for the skull and divided by the $\sqrt{f_0}$? This is then MI_{tc}. If it is MI_{tc}, then it should be labelled MI_{tc}.

Response: Yes, we used the MI_{tc} and labelled it.

We have revised the Results, p7

“The transcranial mechanical index (MI_{tc}; max=1.33) and temperature rise (max at target=0.567°C; max at tissue=2.88°C; max at skull=4.69°C) for both regions remained below the safety limits recommended by the United States Food and Drug Administration (US FDA) (see details in Methods).”

- Report where ISPPA and MI values are computed (target MNI coordinate vs. ROI peak).

Response: Please, see the reply above.

2. Task design and execution

- The semantic task parameters inside the scanner (duration, jittering strategy, timing of stimuli) should be fully described in both the Methods and figure legends.

Response: We have revised the semantic task parameters during the scanning in the Material and Methods and Figure 1.

Materials and Methods, Experimental design and procedure, p19

“For fMRI, participants completed the semantic and control tasks in a block design. Each task block consisted of four trials, and eleven blocks of each task were alternated (e.g., A-B-A-B) with a 4000 ms fixation period between blocks. Each trial began with a 500 ms fixation period, followed by a 3500 ms presentation of the stimuli. Each task had 44 trials, and the total fMRI task took about 7 minutes 30s.”

REDACTED

Figure. 1 A) Experimental design and procedure. B) The block design of fMRI task and an example of semantic association task (left) and control task (right: pattern matching). Each task block consisted of four trials, and eleven blocks of each task were alternated (e.g., A-B-A-B) with a 4000ms fixation period between blocks. Each trial begins with a fixation cross, followed by a stimulus containing three items, a target at the top and two choices at the bottom. In the semantic association task, the correct answer is "milk," as it is meaningfully related to "cow." C) A representative MRS spectrum with estimated peaks. The red line represents the raw spectra, and the blue line shows the post-processed spectra. Within the dotted box, the red line illustrates the estimated model fit for GABA+ and Glx. The black line indicates residuals. D) Diagram of the voxel-based morphometry (VBM) analysis (see the Materials and Methods, VBM analysis).

- Please clarify if the task design was optimized using established jittering procedures (e.g., optseq). It doesn't look like it is but that would be problematic for event related design unless it's a block design.

Response: We employed the block design of fMRI. Please, see our response above.

- Indicate total task time and block structure in the figures for clarity.

Response: We employed the block design of fMRI. Please, see our response above.

- Indicate how VBM was performed in the figure too.

Response: In response to R1's suggestion, we have added a VBM processing workflow diagram to Figure 1. Please, see the reply above.

We have revised the Materials and Methods, VBM analysis, p21

“Data were acquired from two timepoints (ATL and ventricle stimulation sessions) for each subject. To reduce intra-subject variability and improve registration accuracy, an unbiased within-subject average image was generated from both timepoints. Subsequent segmentation and normalization were performed on this average image, ensuring longitudinal consistency in tissue classification. The normalization process used an optimized DARTEL-based approach, estimating high-dimensional warping parameters from the average image and applying them to each individual scan. This approach improved alignment of individual GM and WM maps to a customized template in standard MNI space (Good et al., 2000). Images were corrected for bias field inhomogeneities and segmented into GM, WM, cerebrospinal fluid (CSF), and non-brain tissues. To account for local volume changes, Jacobian modulation was applied, preserving the absolute GM and WM volume at each voxel (Ashburner and Friston, 2000).”

3. fMRI and ROI analyses

- fMRI preprocessing and statistical thresholds are underreported. What Z or T threshold was used before correction?

Response: We apologised that our description of fMRI analysis was not sufficient. We have applied $T > 5$ before the cluster correction.

Accordingly, we have revised the Materials and Methods, fMRI analysis, p22.

“A standard voxel-wise analysis with cluster-level inference was performed, applying a voxel-wise threshold of $p < 0.001$, corresponding to $T > 5$.”

- The use of spherical peak-based ROIs is problematic in the context of highly focal stimulation. I suggest basing ROI selection on the acoustic field with a minimum pressure threshold of 150 or 200 kPa.

Response: We appreciate the reviewer’s suggestion regarding ROI selection based on the acoustic field threshold. While this approach is conceptually appealing, it is not feasible in the current study for several reasons. First, it is unclear whether such a threshold should be defined at the group level or for each participant individually. A group-based threshold would inevitably result in some participants having ROIs above the pressure cutoff while others fall below it. Conversely, participant-specific thresholds would produce ROIs in different anatomical locations across individuals, making group-level comparisons problematic.

Second, threshold-based acoustic ROIs would vary considerably in size and shape (e.g., elliptical for the ATL versus spherical for other regions), introducing further variability in voxel count and interpretability across conditions. By contrast, our use of spherical, peak-based ROIs provides a consistent and comparable approach across participants and regions. Importantly, these spheres encompass a substantial portion of the ATL, ensuring that stimulation reaches a significant, though not identical, subregion in each participant. This approach therefore accommodates inter-individual variability while maintaining methodological consistency, which is crucial for assessing how ATL stimulation impacts system-level function.

Finally, to address the reviewer’s concern regarding the focality of stimulation, we conducted an additional analysis using reduced spherical ROIs of 5 mm radius (Note that fMRI data were smoothed with a 5 mm FWHM kernel) (see the figure below). The results aligned with our original findings, confirming that ATL TUS modulates BOLD activity at the targeted region and exerts effects on both the target and neighbouring areas. These supplementary results further support the robustness of our results.

ROI analysis results in the left ATL.

4. MRS and statistical interpretation

All MRS results need correction for multiple comparisons.

Response: We applied false discovery rate (FDR) correction for multiple comparisons across all MRS results. The corrected outcomes remained consistent with the original findings.

ATL stimulation	FDR-corrected p
GABA	0.049
Glx	0.025
NAA	0.028
Cr	0.028
Cho	0.025

Please provide spectra for all excluded MRS datasets in the supplement.

Response: We have added the excluded MRS spectra as Supplementary Figure 5.

Figure S5. The excluded MRS data. The red line represents the raw spectra, and the blue line shows the post-processed spectra.

Include a collinearity matrix for all MRS measures and explore relationships between MRS metrics and behavior (e.g., correlations or regression models).

Response: We conducted multiple regression analyses using all MRS measures (GABA, Glx, NAA, Cr and Cho) to predict semantic task performance (accuracy [ACC] and reaction time [RT]). None of the models reached statistical significance for either stimulation or behavioural outcomes (all ps > 0.05; see Table below).

ATL stimulation					
ACC			RT		
adjusted R 2	F	p	adjusted R 2	F	p
0.216	1.991	0.147	-0.327	0.211	0.951
Ventricle stimulation					
ACC			RT		
adjusted R 2	F	p	adjusted R 2	F	p
0.183	1.808	0.18	0.008	1.024	0.449

A collinearity matrix was also computed for all MRS variables, showing no evidence of multicollinearity (all tolerance values > 0.2, all VIFs < 10).

ATL stimulation				
	ACC		RT	
ATL VOI	Collinearity Tolerance	Statistics VIF	Collinearity Tolerance	Statistics VIF
GABA	0.84	1.19	0.84	1.19
Glx	0.52	1.92	0.52	1.92
NAA	0.21	4.78	0.21	4.78
Cr	0.19	5.25	0.19	5.25
Cho	0.35	2.87	0.35	2.87
Ventricle stimulation				
	ACC		RT	
ATL VOI	Collinearity Tolerance	Statistics VIF	Collinearity Tolerance	Statistics VIF
GABA	0.38	2.67	0.38	2.67
Glx	0.49	2.06	0.49	2.06
NAA	0.21	4.85	0.21	4.85
Cr	0.23	4.36	0.23	4.36
Cho	0.20	5.04	0.20	5.04

Report behavioural correlations with MRS ratios (e.g., GABA/Glx) and ISPPA/pressure at the MRS voxel.

Response: We performed the correlation analysis between MRS measurement, behaviour and ISPPA/pressure. We found that only ATL Glx/GABA ratio was significantly correlated with semantic task RT following the ATL sonication ($r = 0.5$, $p = 0.03$). Note that Glx/GABA ratio

was significantly higher in the ATL tbTUS compared to the ventricle tbTUS ($t = 7.14$, $p < 0.001$).

We have added these findings in the Results, p10.

“To examine the excitation-inhibition balance (EIB), we computed Glx/GABA ratio and performed the paired t-test. ATL tbTUS significantly increased EIB compared to the control stimulation ($t = 7.14$, $p < 0.001$) (Fig. 5A). There was no difference between the stimulations in the OCC VOI (Fig. 5B). Importantly, the ATL EIB was significantly correlated with semantic task performance (RT: $r = 0.5$, $p = 0.03$) (Fig. 5C).

Figure 5. The results of MRS analysis. A) Comparison of EIB between ATL and ventricle (control) stimulation in the ATL VOI. B) Comparison of EIB between ATL and ventricle (control) stimulation in the OCC VOI. C) the relationship between the ATL EIB and semantic task performance (RT). Grey circles represent individual data. Error bars represent standard error. *** $p < 0.001$, * $p < 0.05$

Consider whether GLMs accounting for session order effects (e.g., ATL first vs. ventricle first) would be more appropriate than t-tests.

Response: We appreciate R1’s comments. However, the study design was fully counterbalanced to control for session order effects (i.e., ATL-first vs. ventricle-first). To formally assess whether order influenced the results, we conducted a GLM including session order as a within-subject factor. This analysis revealed no significant main effects of order (all $ps > 0.356$) and no significant interactions between order and stimulation (all $ps > 0.144$) across neurometabolites, suggesting that session order did not affect the outcomes.

5. Excitatory/inhibitory framing

The framing of tbTUS as “excitatory” needs refinement. There are examples of the same 5 Hz protocol yielding inhibitory responses depending on context and state. A more balanced introduction and discussion, acknowledging variability in outcomes, is essential.

Response: We agree that the characterization of tbTUS as strictly “excitatory” may be overly simplistic. We have revised the Introduction to reflect this, acknowledging that theta-burst TUS may produce either facilitatory or inhibitory effects depending on protocol parameters, brain state, and context.

Introduction, p4

“A theta-burst protocol for transcranial focused ultrasound stimulation (tbTUS) has often been associated with excitatory effects⁴³⁻⁴⁵. However, accumulating evidence indicates that tbTUS can produce both facilitatory and inhibitory outcomes depending on stimulation parameters, brain state, and experimental context^{44, 46}.”

6. The Introduction should include a clear justification for targeting the left ATL specifically. While the ATL's role in semantic memory is well supported, its bilateral involvement is well-documented, and the rationale for left-lateralized stimulation is missing. This omission limits the reader's ability to interpret whether the observed effects are due to lateralized processing or methodological constraints. It should be made clear what could happen with bilateral stimulation.

Response: We appreciate the reviewer’s constructive comment regarding the focus on the left ATL rather than bilateral ATL in our study. Evidence suggests that bilateral ATL systems contribute to semantic representation (for a review, see Lambon Ralph., 2017). Consistent with this, our semantic task induced bilateral ATL activation. Potentially, stimulating both left and right ATL could provide a more comprehensive understanding of tbTUS effects on semantic memory.

However, previous rTMS studies have applied inhibitory stimulation to the left vs. right ATL, demonstrating that stimulation at either site significantly disrupted semantic task performance (Pobric et al., 2007, PNAS; Pobric et al., 2010, Neuropsychologia; Lambon Ralph et al., 2009, Cerebral Cortex). Importantly, these studies reported no significant difference in rTMS effects between left and right ATL stimulation, suggesting that *stimulating either hemisphere produces comparable effects on semantic processing*. In the current study, we combined tbTUS with multimodal imaging to investigate its effects on the ATL. Given our study design constraints (including the need for a control site and control task) and limitations in scanning time, we selected the left ATL as the target region. This choice also aligned with the MRS voxel placement used in our previous studies (Jung et al., 2017; 2025), allowing us to further investigate the underpinning neurochemistry in the ATL. Accordingly, tbTUS was applied to the peak coordinate of the left ventromedial ATL (MNI -36, -15, -30) as identified by previous fMRI studies (Binney et al., 2010; Visser et al., 2012).

Accordingly, we have added the following paragraph in the Introduction

Introduction, p4

“While both ATLs contribute to semantic memory ⁶, we selected the left ATL as our target region. This decision is supported by rTMS findings demonstrating that stimulation of either ATL yields comparable effects on semantic processing ^{14, 15, 49}.”

7. Results: The DCM and VBM analyses should be re-evaluated with ROIs grounded in the simulated acoustic field rather than anatomical peaks.

Response: Please, see our reply above.

Minor comments

- Clarify whether the study was single- or double-blinded for both participants and experimenters. I assume neither, but this should be clear upfront (in the abstract)

Response: The study was conducted in a single-blind manner: participants were informed that stimulation would be applied to different areas of the semantic network, but experimenters were aware of the stimulation site. We have clarified this in the revised abstract.

- The claim: “To our knowledge, this is the first study to demonstrate effective connectivity changes...” is inaccurate and should be removed or revised.

Response: We have removed it.

- The term “theta-burst transcranial ultrasound stimulation (tbTUS)” is misleading. As ITRUSST discourages the term “burst” for pulse trains, consider using “5Hz patterned stimulation” instead.

Response: We appreciate the reviewer’s comment regarding terminology. As the initial study describing this protocol referred to it as “theta-burst transcranial ultrasound stimulation (tbTUS),” subsequent studies have also adopted this terminology and we use identical parameters here, we have retained this terminology for consistency and clarity when comparing across studies. At the same time, we now also describe the protocol as a 5 Hz patterned stimulation to align with current ITRUSST recommendations.

- Clarify whether group-level acoustic parameters (e.g., 191.9 ± 28.06 kPa, ISPPA 1.25 ± 0.4 W/cm²) are computed ROI-wise or are derived from peak locations. As currently stated, they lack interpretability. The mean intensity and MI_{tc} are not very meaningful at the group level, unless the authors can make a claim about these. I would remove and instead focus on interindividual variability

Response: Group-level values were derived from the peak location of the delivered energy. Although mean intensity and MI_{tc} measures are less informative at the group level, we present the averages to provide general context.

- The legend of Figure 2 incorrectly describes acoustic values; please double-check the stated pressure ranges.

Response: We have revised the Figure 2 with the correct acoustic values.

- Improve the figure labelling to include task design details, stimulus timing, and data collection timelines.

Response: We have revised the Figure 1.

- The VBM section has a grammar issue: “A statistical threshold of $p < 0.005$ (uncorrected).” Is not a sentence.

Response: We have revised the sentence.

Reviewer #2 (Remarks to the Author):

This study presents evidence of the neuromodulatory effects of theta-burst transcranial ultrasound stimulation (tbTUS) on semantic memory, utilizing multimodal neuroimaging to reveal structural, biochemical, and functional changes. While the findings are promising, several key issues need further clarification to enhance the manuscript's impact and ensure its reproducibility.

Response: We appreciate R2's positive evaluation of our manuscript.

Major comments

1. The study enrolled 23 participants, but the final analyses included only 21 (after exclusions). Given the complexity of multimodal imaging and the subtlety of neurochemical/functional changes, the sample size raises concerns about statistical power, particularly for subgroup analyses (e.g., MRS exclusions). Provide a formal power analysis to justify the sample size, especially for detecting interactions (e.g., stimulation × session effects).

Response: Our sample size was determined a priori using data and effect sizes from TMS studies with comparable designs and outcome measures (Jung & Lambon Ralph., 2016; 2021; Jung et al., 2022; 2025).

In order to provide robust power calculations for each method we have: (a) utilised data from previous whole experiments; (b) selected experiments which have the same design; (c) for the fMRI analysis, we picked a critical ROI from within the semantic network with the weakest intrinsic fMRI signal (ATL), to be maximally conservative.

fMRI: power analysis was calculated based on Mumford & Nichols's fMRI power toolbox (<http://fmripower.org/>).

Semantic vs. control task from the fMRI-TMS data block-fMRI design with N=23 participants for $\alpha=0.05$, power=80%, interaction TMS (ATL vs OCC) x task effect size = 0.33 in behaviour and fMRI signal, then $N > 16$ are required.

TMS, TMS/fMRI and TMS/fMRI/MRS: The study used a 2 (ATL target vs. control site) × 2 (semantic vs. control task) x session (PRE vs POST) within subject design. The data from N from 21 to 23 participants indicate that to achieve $\alpha=0.05$, power=80% for the critical interaction (effect size ≈ 0.35) in MRS, behavioural and fMRI measurement then $N > 18$ are required.

2. Multiple researches have reported that different parameter settings of TUS can lead to varying neurofunctional regulatory effects (excitatory or inhibitor). More information is needed on how the tbTUS stimulation parameters were chosen in this study and whether they were optimized for the target of ATL and the specific cognitive task. The relationship between parameters and effects should be mentioned in the discussion.

Response: We thank the reviewer for this helpful comment and agree that clarification of the stimulation parameters is important. In this study, we employed the tbTUS protocol as defined in prior work (Zeng et al., 2022, *Ann Neurol*). Zeng et al. (2022) reported that this protocol increased motor-evoked potentials, reduced short-interval intracortical inhibition,

enhanced intracortical facilitation, and improved motor performance (faster RTs). More recently, Yaakub et al. (2023) applied the same protocol to the posterior cingulate cortex (PCC), demonstrating reduced GABA levels at the stimulation site and associated changes in functional connectivity (FC) within the PCC network. Atkinson et al. (2024) also used this protocol in the inferior frontal gyrus (IFG), reporting faster reaction times on the stop-signal task along with IFG FC changes. Although these studies did not target the ATL directly, they provide convergent evidence that this protocol can reliably induce neurophysiological, neurochemical and behavioural effects.

We have revised the Discussion.

Discussion, p16

“There are several limitations in this study. First, our sample size is relatively small. Second, while we employed a tbTUS protocol that has frequently been associated with excitatory effects^{39, 41, 89}, accumulating evidence indicates that tbTUS can yield either facilitatory or inhibitory outcomes depending on brain state, and experimental context^{44, 46}. In this study, we demonstrated the facilitatory effects across behaviour, neurochemistry and fMRI measures, but further research is needed to validate and extend these findings. Third, MRS can detect the total concentration of GABA and Glx within a voxel of interest but cannot differentiate between distinct pools of these neurochemicals⁹⁰. In addition, the MRS-derived Glx signal reflects the combined tissue content of glutamate and glutamine, without distinguishing between the two⁶⁴. Despite these limitations, linking GABA/Glx shifts with functional connectivity offers important insights into ultrasound-induced plasticity. Fourth, although ventricle-TUS served as an active control condition in this study and our previous work has validated its suitability by showing that it preserves the auditory and somatosensory features of active stimulation without altering brain structure or function⁴⁸, we acknowledge that additional controls (e.g., sham or off-target stimulation) would further strengthen the specificity of the conclusions. Finally, to validate our findings, future work should include longitudinal studies with long-term interventions to establish the persistence of ATL tbTUS effects.”

3. Ventricle stimulation was used as a control site. While prior work supports this approach (Atkinson-Clement et al., 2024), the manuscript does not fully address whether ventricular stimulation itself may influence cerebrospinal fluid dynamics or adjacent neural structures (e.g., thalamocortical pathways). Include additional controls (e.g., sham stimulation or off-target stimulation) to rule out nonspecific effects of ultrasound energy propagation will enhance the credibility of the conclusions.

Response: We thank the reviewer for raising this important point. In our previous work, we directly compared sham with auditory masking, ventricle stimulation, and ATL stimulation to evaluate the suitability of ventricle-TUS as a control condition. These analyses demonstrated that ventricle-TUS preserves the auditory, somatosensory, and procedural experiential features associated with active TUS, yet does not produce measurable changes in brain structure or function. This conclusion was supported across multiple neuroimaging metrics, including whole-brain analyses, network-based measures, and ROI-level assessments within the lateral ventricles.

We therefore consider ventricle-TUS a valuable addition to the set of available control conditions, as it addresses limitations of traditional approaches. For example, although auditory masking with headphones (sham) has been used to minimize sound-related confounds, studies have shown that masking or ramping alone cannot reliably obscure the audible noise generated during TUS. More recently, Kop et al. (2022) highlighted the limitations of relying solely on flip-over sham stimulation without an active control. In this context, ventricle-TUS offers a simple and effective active control that matches the auditory and somatosensory characteristics of ATL-TUS without driving neural effects.

We acknowledge it as the limitation in the Discussion. While our results and prior validation strongly support the use of ventricle stimulation, future studies could incorporate additional controls (e.g., sham or off-target stimulation) to further strengthen the conclusions.

4. MRS findings (e.g., decreased GABA+/increased Glx) are interpreted as shifts in excitation-inhibition balance. However, MRS cannot distinguish synaptic vs. extrasynaptic GABA/glutamate pools or differentiate glutamine from glutamate contributions. I suggest the authors discuss these limitations. Linking neurochemical shifts (GABA/Glx) to functional connectivity changes will advance the understanding of ultrasound-induced plasticity.

Response: We have revised the Discussion to acknowledge these limitations explicitly. Please, see our reply to R1 above.

Discussion, p14

“MRS-measured GABA has been suggested to primarily reflect tonic inhibition rather than synaptic GABAergic signalling^{59, 60}, though evidence remains inconsistent^{61, 62}. A recent study identified a link between MRS-detected GABA and phasic synaptic activity⁶³, suggesting that it may capture both tonic and synaptic GABAergic transmission. Similarly, MRS-Glx measures total glutamate and glutamine concentrations but cannot reliably distinguish between them. MRS-detectable glutamate is closely linked to energy metabolism, as it serves as a key metabolite in the tricarboxylic acid (TCA) cycle^{64, 65}. Neuronal activity and metabolism are highly interconnected, with glucose consumption and glutamate-glutamine cycling increasing in response to task-related activation^{64, 66}. Studies combining MRS and fMRI have shown that GABA and Glx concentrations across cortical regions are associated with stimulus-induced BOLD signal changes and functional connectivity for a review, see⁶⁷.”

5. The reported grey matter (GM) volume increases in the ATL after a single tbTUS session are striking. VBM is sensitive to hydration, blood flow, and registration errors. The manuscript does not rule out transient hemodynamic or fluid shifts as contributors. Include longitudinal follow-up scans to assess the persistence of GM changes and correlate them with behavioral outcomes.

Response: We appreciate the reviewer’s insightful comment. As noted in the revised Discussion, longitudinal follow-up scans would be an ideal next step with long-term intervention to validate our findings. Please, see our reply above.

6. The manuscript mentions the potential role of tbTUS in enhancing cognitive function, but the long-term effects of this intervention are not adequately discussed.

Response: Please, see our reply above.

Minor comments

1. Specify the criteria for MRS data exclusion (e.g., linewidth thresholds) in the main text, not just supplementary materials.

Response: Thank you. We report this information in the Methods and Materials, MRS analysis, p12

“Data quality was assessed for each voxel using metrics such as fit errors, signal-to-noise ratio (SNR), and linewidth. Data exclusion criteria included fit errors > 15% for each neurochemical, water linewidths (FWHM) > 20 Hz, and SNR < 40.”

2. The semantic task lacks difficulty titration. Variability in baseline performance (Fig. S1) suggests floor effects.

Response: The semantic task used in this study has been extensively validated in both healthy and clinical populations, including patients with dementia and post-stroke semantic impairments, and is widely employed in dementia clinics. Moreover, our previous work has shown that this task reliably captures individual differences in semantic memory, linking performance to underlying neurochemistry (Jung et al., 2017; 2025) and responsiveness to TMS (Jung et al., 2021; 2025).

3. Consider adding a schematic summarizing the proposed mechanism (Fig. 6 is dense).

Response: Thank you for this suggestion. After careful and long deliberation, we have found that it is impossible to capture the various mechanistic hypotheses in a clear manner. Figure 6 merely summarises the DCM results.

Reviewer #3 (Remarks to the Author):

This is an impressive set of results from a complex experiment that investigates effects of tbTUS on semantic processing networks across behaviour and multiple imaging modalities. The manuscript is well written. The authors demonstrate that tbTUS to the left anterior temporal lobe has a positive effect on semantic processing task performance, alongside increasing grey matter volume, modulating neurochemical markers and altering task-related BOLD signal across semantic brain networks.

Response: We appreciate R3's supportive feedback on the manuscript.

Major

Behavioural task

More detail is needed about the behavioural tasks. For example: How many trials/blocks were there? Was the paradigm for the PRE and POST sessions the same as the in-scanner session (e.g. number of trials)? Were the stimuli the same or different for each session? Were participants instructed to respond as rapidly as possible?

If the stimuli used were the same each time the task is performed, is it possible that stimulation was acting through some mechanism other than directly improving semantic associative processing, but instead, for example, episodic learning over sessions?

Response: We apologise the lack of details of the task. Importantly, stimuli were not repeated across sessions in order to avoid learning-related effects.

We have revised the Methods and Materials, Experimental design and procedure, p19

“Each task consisted of 116 trials. Participants were instructed to respond as quickly and accurately as possible by pressing a key. Stimuli were not repeated across the pre- and post-stimulation sessions.”

“For fMRI, participants completed the semantic and control tasks in a block design. Each task block consisted of four trials, and eleven blocks of each task were alternated (e.g., A-B-A-B) with a 4000ms fixation period between blocks. Each trial began with a 500 ms fixation period, followed by a 3500 ms presentation of the stimuli. Each task had 44 trials, and the total fMRI task took about 7 minutes 30s”

An increase with TUS of ~4% accuracy is only about 1 trial better accuracy. How meaningful/robust is this finding? In Supplementary figure 1, it appears that there is one participant who improves much more than others. Is the result still significant if this participant is removed?

Response: The task consisted of 116 trials, so a ~5.7% accuracy increase corresponds to an improvement of approximately 6.6 trials. To our knowledge, this is the first evidence of

accuracy enhancement induced by non-invasive brain stimulation in semantic memory. Prior studies using TMS or tDCS have typically reported effects on reaction times ($\approx 100\text{--}250$ ms) rather than accuracy.

Yes, the improvement observed here is statistically robust: when the participant showing the largest gain was excluded, the results remained significant (PRE: $72.7 \pm 9.0\%$, POST: $75.9 \pm 6.7\%$, $t = -3.56$, $p < 0.001$). Please, see the figure below.

VBM

how was the SVC (MNI: $-36, -15, -30$) chosen? How does it relate to i) the stimulated region and ii) known semantic networks from prior activation (e.g. fMRI/icEEG) studies?

fMRI: What was task performance like in the scanner?

Response: The ATL target was defined using MNI coordinates ($-36, -15, -30$) derived from prior fMRI studies employing the same semantic task^{24,52}. This site corresponds to the ventromedial ATL, a region consistently implicated in semantic representation by convergent evidence from fMRI, TMS, and intracranial EEG and cortical stimulation studies.

Task performance in the scanner is summarised below. Importantly, ATL stimulation significantly improved performance relative to the pre-session baseline, both in accuracy (PRE = 73.2% , $p = 0.009$) and reaction time (PRE RT = 1.1 s, $p < 0.001$). In contrast, ventricle stimulation produced no significant changes compared with the pre-session baseline (accuracy = 75.2% , RT = 1.2 s). It should be noted that each task consisted of only 44 trials.

	Semantic		Control	
	ACC (%)	RT (s)	ACC (%)	RT (s)
ATL stimulation	76.1 ± 8.4	0.9 ± 0.1	88.9 ± 9.1	1.2 ± 0.3
Ventricle stimulation	74.9 ± 5.6	1.1 ± 0.3	89.6 ± 8.4	1.6 ± 0.2

Since the duration of TUS effects on behaviour and brain remains poorly understood, it would be helpful to know if there was any effect of stimulation order on the results? There is no discussion about whether the multiple behavioural and imaging effects of tbTUS were related to each other across participants. The questions are obvious (e.g. did changes in effective connectivity relate to behaviour/MRS/VBM effects?), although there are likely to be power issues. Nevertheless, some mention of how the behavioural and imaging results relate to each other seems important.

Response: The study design was fully counterbalanced to control for session order effects (i.e., ATL-first vs. ventricle-first). To formally assess whether order influenced the results, we conducted a GLM including session order as a within-subject factor. This analysis revealed no significant main effects of order (all p s > 0.356) and no significant interactions between order and stimulation (all p s > 0.144).

We performed the correlation analysis between MRS measurement, behaviour, connectivity, and ISPPA/pressure. We found that only ATL Glx/GABA ratio was significantly correlated with semantic task RT following the ATL sonication ($r = 0.5$, $p = 0.03$). Please, see our reply to R1.

The Discussion should include a section exploring potential weaknesses of the study.

Response: We have added the limitation section in the Discussion.

Minor

Response: Thank you for R3's careful comments. We have addressed all of them in the revision.

line 118: Typo "Combined with MRS enables..."

line 158: "Left ventricle" should presumably read "Left lateral ventricle"

Results section: In the first paragraph it would help to clarify that scanning took place in both session 2 and session 3 and this should also be added to Fig 1a.

Figure 1a) This could be clearer in general. For example, show that patients were randomised into either the ventricle then ATL group or the ATL then ventricle group.

Behavioural results: The ANOVA should really include factors of stimulation, session AND task.

Response: We thank the reviewer for this valuable suggestion. In line with the comment, we re-ran the behavioural ANOVA including stimulation (ATL vs. ventricle), session (PRE vs. POST), and task (semantic vs. control) as factors. The results (accuracy) revealed a significant main effect of task ($F = 282.89$, $p < 0.001$) and a significant task \times session interaction ($F = 8.46$, $p = 0.008$). The three-way stimulation \times session \times task interaction did not reach significance ($F = 2.29$, $p = 0.14$), likely reflecting the modest sample size. Given

our a priori hypothesis that ATL TUS would selectively affect semantic but not control task performance, we conducted task-specific analyses to detect these subtle stimulation effects on higher cognition.

Figure 4: The caption says that error bars represent standard error. There are no error bars on the figure.

Response: We have removed the sentence.

Figure 6e: The white bars are missing.

Response: The white bar values for the ventricle stimulation condition were extremely small (≈ 0.005) compared to those for ATL stimulation, which is why they are not visible in the bar graph.

REVIEWERS' COMMENTS

Reviewer #1 (Remarks to the Author):

I have now reviewed this manuscript for the second time. The authors have addressed many of my previous comments, and the revisions have substantially improved the clarity and coherence of the paper. The study presents interesting and potentially important findings regarding the effects of TUS on semantic memory, supported by a comprehensive multimodal approach.

However, there remain a few issues that should be addressed before I can endorse the manuscript for publication:

Data availability and documentation

The online dataset is available, but there is no accompanying metadata, readme file, or explanation of the task structure, data organization, or variable definitions. There are also no acquisition parameter summaries or analysis instructions. These materials are essential for reproducibility and transparency. The dataset must be improved before publication to ensure it meets FAIR principles and open-research expectations for findability, accessibility, interoperability, and reusability.

Response: We apologize for the missing data. Due to the large size of the dataset, we encountered difficulties uploading all files to OSF. We will ensure that the complete dataset, along with all required documentation, is fully uploaded prior to publication.

Task validation

Please add a brief statement acknowledging that the semantic association task has been extensively validated in previous studies by the authors' group, but that its generalizability to independent implementations by other laboratories remains to be established. This clarification will strengthen transparency and situate the work appropriately within the broader literature.

Response: Following R1's suggestion, we have added the following sentences to the Methods section:

“Participants performed a semantic association task and pattern matching as a control task prior to tbTUS. These tasks have been extensively validated in both healthy and dementia populations⁹²⁻⁹⁴ and have been previously used to examine the relationship between semantic task performance, neurochemical profiles, and neural activation during semantic processing²⁴⁻²⁶.”

Blinding procedures

The manuscript states that the study was single blind, but it remains unclear how blinding integrity was verified. Please explicitly describe how blinding was checked. If

this was not formally assessed, this limitation should be acknowledged in the discussion.

Response: Participants were informed that stimulation would be applied to different areas of the semantic network, while experimenters were necessarily aware of the stimulation site to ensure accurate targeting. To assess blinding integrity, at the end of all sessions and following a full debriefing, participants were verbally asked whether they could identify which session involved ATL stimulation. None of the participants were able to distinguish between the stimulation conditions.

Accordingly, we have added the following sentences to the Methods section:

“Participants were informed that stimulation would be applied to different areas of the semantic network, although experimenters were aware of the stimulation site to ensure accurate targeting. To check blinding integrity, participants were verbally asked at the end of all sessions after being informed that one session involved control stimulation, whether they could distinguish between the two stimulation conditions. No participants were able to identify the control session.”